



**Spatial and temporal variations in snow chemistry along a traverse**
**from coastal East Antarctica to the ice sheet summit (Dome A)**
Guitao Shi[1,2], Hongmei Ma[2], Zhengyi Hu[2], Zhenlou Chen[1], Chunlei An[2], Su Jiang[2], Yuansheng Li[2],
Tianming Ma[2], Jinhai Yu[2], Danhe Wang[1], Siyu Lu[2], Bo Sun[2], and Meredith G. Hastings[3]
[1] Key Laboratory of Geographic Information Science (Ministry of Education), School of Geographic
Sciences and State Key Lab of Estuarine and Coastal Research, East China Normal University,
Shanghai 200241, China
[2] Polar Research Institute of China, Shanghai 200062, China,
[3] Department of Earth, Environmental and Planetary Sciences and Institute at Brown for Environment
and Society, Brown University, Providence, Rhode Island 02912, USA.
*Correspondence to: G Shi (gtshi@geo.ecnu.edu.cn)





**Abstract**
There is a large variability in environmental conditions across the Antarctic ice sheet, and it is of
significance to investigate the snow chemistry at as many locations as possible and over time, given
that the ice sheet itself, and precipitation and deposition patterns and trends are changing. The China
inland Antarctic traverse from coastal Zhongshan Station to the ice sheet summit (Dome A) covers a
variety of environments, allowing for a vast collection of snow chemistry conditions across East
Antarctica. Surface snow and snow pit samples were collected on this traverse during five campaigns,
to comprehensively investigate the spatial and temporal variations in chemical ions ($Cl^-$, $NO_3^-$, $SO_4^{2-}$,
$Na^+$, $NH_4^+$, $K^+$, $Mg^{2+}$, and $Ca^{2+}$) and the related controlling factors. Results show that spatial patterns of
ions in surface snow are consistent among the five campaigns, with $Cl^-$, $Na^+$, $K^+$, and $Mg^{2+}$ decreasing
rapidly with distance from the coast and $NO_3^-$ showing an opposite pattern. No clear spatial trends in
$SO_4^{2-}$, $NH_4^+$, and $Ca^{2+}$ were found. In the interior areas, an enrichment of $Cl^-$ versus $Na^+$ with respect to
seawater composition is ubiquitous as a result of the deposition of HCl, which can account for up to
~40 % of the total $Cl^-$ budget, while enriched $K^+$ and $Mg^{2+}$ are associated with terrestrial particle mass.
$Ca^{2+}$ and $SO_4^{2-}$ in surface snow are significantly enriched relative to $Na^+$, related to terrestrial dust
inputs and marine biogenic emissions, respectively. Snow $NH_4^+$ is mainly associated with marine
biological activities, with higher concentrations in summer than in winter. On the coast, parts of the
winter snow are characterized with a depletion of $SO_4^{2-}$ versus $Na^+$, and a significant negative
correlation between $nssSO_4^{2-}$ and $Na^+$ was found, suggesting that sea salts originated from the sea ice.
In the interior areas, the negative $nssSO_4^{2-}$ signal in winter snow resulted from inputs of sea salts being
completely swamped by the contribution of marine biogenic emissions. Ternary plots of $Cl^-$, $Na^+$, and
$SO_4^{2-}$ suggest that sea salt modification is generally negligible on the coast, while the degree of
modification processes to sea salts is high in the interior areas, especially during the summertime. Ion
flux assessment suggests an efficient transport of $nssSO_4^{2-}$ to at least as far inland as the ~2800 m
contour line. The interannual variations in ion concentrations in surface snow on the traverse are likely
linked to the changes in the Southern Indian Ocean low (SIOL) from year to year, and the deepening of
the SIOL in summer tends to promote the transport of marine aerosols to Princess Elizabeth Land.




## 1 Introduction

Snow can scavenge the atmospheric chemicals, including sea salts, acids and other organic species,
and thereby ice cores can provide the most direct records of the composition of the atmosphere. In
Antarctica, the EPICA Dome C ice core encompassed more than 800 ka of sequential glaciochemical
data, the longest available records obtained thus far from ice cores (EPICA Community Members, 2004;
Wolff et al., 2006; Jouzel et al., 2007; Kaufmann et al., 2010). Major chemical ions are among the core
classical measurements in snow and ice due to that they are indicative of a wealth of climate
information (e.g., marine biological activity, sea ice extent, and atmospheric circulation pattern). In
comparison with trace gases trapped in ice core bubbles, the accurate interpretation of glaciochemical
records is challenging since chemicals in the ice can be indicative of a combination of sources,
transport strength, and preservation processes. At times, records of a particular species in ice cores can
be interpreted differently amongst sites, e.g., sodium ($Na^+$) variability in ice cores might indicate the
changes in sea ice extent or atmospheric meridional transport strength at varied locations (Goodwin et
al., 2004; Severi et al., 2017). Therefore, there has been great interest in the determination of sources,
mechanisms, and pathways of transport as well as preservation mechanisms of chemicals in snow and
ice (e.g., Mahalinganathan et al., 2011; Dixon et al., 2013; Shi et al., 2018a), and a better understanding
of snow chemistry is crucial towards an accurate interpretation of glaciochemical records from ice
cores.
Snow chemistry has been broadly investigated along traverses during the International
Trans-Antarctic Scientific Expedition (ITASE), e.g., DDU to Dome C, Syowa to Dome F, Terra Nova
Bay to Dome C, 1990 ITASE, and US ITASE in West Antarctica (Qin et al., 1992; Mulvaney and Wolff,
1994; Proposito et al., 2002; Suzuki et al., 2002; Dixon et al., 2013), and Bertler et al. (2005) has
comprehensively summarized the glaciochemical data across the ice sheet, most of which are for
surface snow. Among the major ions, sea salt related ions (e.g., $Na^+$ and $Cl^-$), in general, are the most
abundant species, and typically exhibit a clear spatial trend, with concentrations falling off sharply with
distance from the coast. Sea salts in snow are traditionally thought to be from sea spray in open water,
and higher wind speeds (more efficient production and transport) are proposed to be responsible for the
higher sea salt concentrations in winter (Delmas, 1992). Recently, sea salt aerosols produced from
blowing snow above sea ice is thought to be a major source of sea salt related ions (Wolff et al., 2003;
Frey et al., 2020), and this tends to well explain their higher concentrations in the winter snow. Also, it
is proposed that sea salt aerosols originated from sea ice can be efficiently transported to central
Antarctica (Udisti et al., 2012; Legrand et al., 2017b), and thus sea salts can be a proxy in ice cores for
sea ice coverage (Abram et al., 2013 and references therein). Acidic ions such as nitrate and sulfate
($NO_3^-$ and $SO_4^{2-}$) are typically also abundant ionic species in snow, both of which can be deposited as
salts in aerosols, and as gaseous acids. $SO_4^{2-}$ in the snow is mainly from marine biogenic sulfur species,
dimethylsulphide (DMS) (Saltzman, 1995), with a small proportion from sea salt aerosols, while large
volcanic eruption emissions can episodically contribute to spikes in $SO_4^{2-}$ concentration (Jiang et al.,
2012; Cole-Dai et al., 2013). Thus, $SO_4^{2-}$ in ice cores can be indicative of ocean productivity in the past
(e.g., Wolff et al., 2006). Sources of $NO_3^-$ are sometimes complicated to identify, due to the
post-depositional processing after deposition into the snowpack (e.g., photolysis and volatilization),
and stratospheric input and tropospheric transport from mid-low latitudes have been proposed to be
important sources (Wagenbach et al., 1998b; Savarino et al., 2007; Lee et al., 2014; Shi et al., 2015;
Shi et al., 2018a). As for calcium ($Ca^{2+}$) in snow, both long range transport of terrestrial particle mass
and sea salt aerosols are important sources, and $Ca^{2+}$ in ice cores recovered from interior areas is more



likely associated with terrestrial inputs (e.g., Wolff et al., 2006). Terrestrial sources can also contribute
to potassium ($K^+$) and magnesium ($Mg^{2+}$) in snow, but the contribution proportion varies significantly
among sites (Keene et al., 2007; Khodzher et al., 2014). In comparison with the other species,
ammonium ($NH_4^+$) in the snow has been rarely investigated due to the low concentration, and biogenic
emissions in the Southern Ocean and/or mid-latitude biomass burning were proposed to be the major
sources, depending on the investigation sites (Kaufmann et al., 2010; Pasteris et al., 2014). In summary,
source identification of ions in Antarctic snow and ice has been conducted intensely, however, the site-
and area-specific investigations are needed.

95        With varied sources and lifetimes, ions in snow often exhibit different seasonal variations, e.g., sea
salt related ions show high concentrations in winter, while elevated concentrations of $SO_4^{2-}$ and $NO_3^-$
are frequently observed in summer (e.g., Wagenbach, 1996; Gragnani et al., 1998; Traversi et al., 2004;
Shi et al., 2015). Indeed, these ions are frequently taken as seasonal markers for snow pit and ice core
dating. On annual to decadal time scales, ion concentrations in snow and ice tend to be associated with
changes in transport from year to year (Xiao et al., 2004; Severi et al., 2009), and thus large scale
atmospheric and oceanic circulation in the Southern Hemisphere, such as the Southern Annular Mode
(SAM), Southern Oscillation (SO) and Southern Indian Ocean Dipole (SIOD), could potentially
influence variations in ions in ice (Russell and McGregor, 2010; Mayewski et al., 2017). For instance,
the variability of $Na^+$ in the Law Dome ice core was mostly like associated with interannual changes in
SAM that dominates the meridional aerosol transport from mid-latitude sources (Goodwin et al., 2004).
In addition, sea ice coverage around Antarctica plays an important role in variations in ions, and larger
sea ice coverage is linked with higher sea salt concentrations, as well as non-sea salt $SO_4^{2-}$ ($nssSO_4^{2-}$)
concentrations in ice, particularly over glacial-interglacial time scales (Kaufmann et al., 2010; Wolff et
al., 2010; Abram et al., 2013). In addition to sources, lifetime, and transport processes, the preservation
of ions is an important factor influencing concentrations in snow and ice, particularly the volatile
species (e.g., $NO_3^-$ and $Cl^-$). Post-depositional processes can result in significant losses of volatile
species in snow, particularly at sites with low snow accumulation rate (e.g., East Antarctic plateau)
(Wagnon et al., 1999; Sato et al., 2008; Shi et al., 2015). In summary, spatial and temporal variations in
snow chemistry are influenced by a variety of factors, and further observations of ions in snow are
needed to determine the controlling factors for particular times and places.

116       Although investigations of snow chemistry have been carried out along several overland traverses,
many Antarctic areas remain undocumented. In addition, the Antarctic ice sheet itself, and precipitation
and deposition patterns and trends are changing, and the investigation of snow chemistry under
different environmental conditions and over time is needed. The China inland Antarctic traverse from
coastal Zhongshan Station to the ice sheet summit (Dome A) covers a distance of 1256 km in the
Indian Ocean sector. The first China inland Antarctic expedition took place in 1999, reaching the site
~300 km from the coast, and in 2005, this traverse extended to Dome A plateau (with elevation ~4100
m), where the oldest ice (~one million-year old) was thought to be preserved (Zhao et al., 2018). This
traverse covers a range of environments, e.g., high snow accumulation rate is present on the coast and
in some interior areas, and very low accumulation rate is observed on the Dome A plateau. It is noted
that some of the interior areas are greatly influenced by persistent wind scour, leading to near zero
snow accumulation (Das et al., 2013; Ding et al., 2015). This traverse, thus, provides further
opportunity to investigate snow chemistry and its main controlling factors in different environments. In
addition, the Dome A deep ice core reached a depth of 803 m in 2019, and an investigation of snow
chemistry in the Indian Ocean sector, especially on the Dome A plateau would be of significance to the



interpretation of the deep ice core. Several investigations have been carried out in the past to determine
the concentrations and spatial patterns of a few ionic species and trace elements on the traverse (e.g., Li
et al., 2016; Du et al., 2019), but limited snow chemistry data were previously available. Additionally,
the interannual variations in snow chemistry and the related controlling factors on the traverse are far
from understood. Therefore, we used surface snow and snow pit samples collected during five China
inland Antarctic scientific expedition campaigns, to determine the spatial and temporal variations in a
comprehensive set of ions ($Na^+$, $NH_4^+$, $K^+$, $Mg^{2+}$, $Ca^{2+}$, $Cl^-$, $NO_3^-$, and $SO_4^{2-}$) and their controlling
factors. This work also presents data on snow chemistry from a less documented area, particularly the
Dome A area, providing baseline values of snow ions and records of significance for evaluating
potential changes in atmospheric chemistry over Antarctica under a warming climate.

**2 Methods**
**2.1 Sample collection**
Snow samples were collected along the traverse from the coast to the ice sheet summit during five
Chinese National Antarctic Research Expedition (CHINARE) campaigns (Fig. 1). In January 1999, 107
surface snow samples were collected on the traverse (from coast to the site ~1100 km from the coast;
the Chinese inland traverse coverage did not extend to Dome A then). In January and February in the
years 2011, 2013, 2015, and 2016, 120, 125, 117, and 125 surface snow samples were collected on the
traverse, respectively. In total, 594 snow samples were collected during the five seasons.
Surface snow samples were collected at ~10 km intervals, and the sampling sites are generally >500
m away from the traverse route to avoid possible contamination from expedition team activities.
During snow sampling, all personnel wore polyethylene (PE) gloves and face masks, and pushed the
high-density polyethylene (HDPE) bottles horizontally into the surface snow layer (~3 cm) in the
windward direction.
In addition to surface snow, snow pits were sampled in three representative areas on the traverse (P1,
P2, and P3; Fig. 1). P1, located on the coast (76.49 °E, 69.79 °S; 46 km from the coast), was sampled in
December 2015; P2, located in the interior area (77.03 °E, 76.42 °S; 800 km from the coast), was
sampled in January 2016; P3, located on Dome A plateau (77.11 °E, 80.42 °S; 1256 km from the coast),
was sampled in January 2010. Sites P1 and P2 are characterized with high snow accumulation rate
(>100 kg m$^{-2}$ a$^{-1}$), while snow accumulation rate at P3 is ~25 kg m$^{-2}$ a$^{-1}$. The depths of P1, P2, and P3
are 180, 100, and 150 cm, respectively, with the respective sampling resolution of 5, 3, and 1 cm. Snow
pit samples were collected using the narrow mouth HDPE bottles pushed horizontally into the snow
wall from the bottom of the pit and moving upwards.
All of the bottles used for snow sampling were pre-cleaned with Milli-Q water (18.2 MΩ), dried in a
class 100 super clean hood and then sealed in clean PE bags that were not opened until the field
sampling started. During each sampling campaign, three pre-cleaned bottles filled with Milli-Q water
taken to the field and treated to the same conditions as field samples represent field blanks. After
collection, the bottles were again sealed in clean PE bags and preserved in clean expanded
polypropylene boxes. All samples were transported and stored under freezing conditions (~-20 °C).

**2.2 Sample analysis**
Snow samples were first melted in the closed bottles on a super clean bench (class 100) before
chemical measurements. In the class 100 room, about 5 ml of the melted sample was transferred to the
pre-cleaned 8-ml ion chromatography (IC) autosampler vials, and then the lid was tightly screwed on to



the vials. The samples were analyzed by IC for the concentrations of ions ($Na^+$, $NH_4^+$, $K^+$, $Mg^{2+}$, $Ca^{2+}$,
$Cl^-$, $NO_3^-$, and $SO_4^{2-}$). (Note that the IC was installed in a class 1000 clean room) The samples collected
in 1999 were analyzed by using the DX-500 IC system (Dionex, USA), while the snow collected in the
other campaigns were analyzed using an ICS-3000 IC system (Dionex, USA). The eluents for cations
and anions were methanesulfonic acid (MSA) and potassium hydroxide (KOH), respectively. More
details on this method are described in Shi et al. (2012). During sample analysis, replicate
determinations ($n = 5$) were performed, and one relative standard deviation ($1\sigma$) for all eight ions was
generally <5 %. In addition, the pooled standard deviation of all replicate samples run in at least two
different sets was examined ($n = 65$) and yielded 0.020, 0.023, 0.038, 0.022, 0.039, 0.005, 0.008, and
0.005 µeq $L^{-1}$ for $Cl^-$, $NO_3^-$, $SO_4^{2-}$, $Na^+$, $NH_4^+$, $K^+$, $Mg^{2+}$, and $Ca^{2+}$, respectively. Ion concentrations in
field blanks are lower than the detection limit (DL, 3 standard deviations of water blank in the
laboratory).
In Antarctic snow, concentrations of $H^+$ are usually not measured directly, but deduced from the
ion-balance disequilibrium in the snow. Here, $H^+$ concentration is calculated as follows.
$[H^+] = [SO_4^{2-}] + [NO_3^-] + [Cl^-] - [Na^+] - [NH_4^+] - [K^+] - [Mg^{2+}] - [Ca^{2+}]$ Eq. (1),
where ion concentrations are in µeq $L^{-1}$. In addition, the non-sea-salt fractions of ions (nssX), including
$nssCl^-$, $nssSO_4^{2-}$, $nssK^+$, $nssMg^{2+}$ and $nssCa^{2+}$, can be calculated from the following expression,
$[nssX] = [X]_{snow} - ([X]/[Na^+])_{seawater} \times [Na^+]_{snow}$ Eq. (2),
where [X] is the concentration of ion X, and $[X]/[Na^+]$ ratios in seawater are 1.17 ($Cl^-$), 0.12 ($SO_4^{2-}$),
0.022 ($K^+$), 0.23 ($Mg^{2+}$) and 0.044 ($Ca^{2+}$) (in µeq $L^{-1}$).

**2.3 Enrichment assessment of ions**
The enrichment factor (EF) is a measurement of whether or not an ion is present in a relative
abundance similar to that of seawater, which can be calculated as follows.
$EF_X = ([X]/[Na^+])_{snow} / ([X]/[Na^+])_{seawater}$ Eq. (3).
When EF > 1.0, the ion X is enriched, i.e., additional sources are present in addition to sea salt spray.
EF <1.0 corresponds to the depletion of ion X, possibly indicating the presence of fractionation.
In both equations (2) and (3), we assume that $Na^+$ is exclusively from the sea spray (i.e., the sea salt
indicator) in surface snow based on the following facts: 1) the $Cl^-/Na^+$ ratios in snow samples are
generally above 1.17, the average value in seawater (Nozaki, 2001), 2) the contribution of dust
leachable Na is negligible in Antarctic snow (Legrand and Delmas, 1988; Röthlisberger et al., 2002),
and 3) negligible $Na^+$ fractionation resulted from mirabilite ($Na_2SO_4 \cdot 10H_2O$) precipitation in sea-ice
formation at <-8°C (Marion et al., 1999), especially considering the smallest sea ice extent in late
summer in East Antarctica (Holland et al., 2014).

**2.4 Principal component analysis (PCA) of ions**
The essence of PCA is converting the observed variables into factors or principal components, so
that a minimized set of underlying variables can be identified. Bartlett sphericity test and
Kaiser-Meyer-Olkin test indicated that the raw data (i.e., ion concentrations in surface snow) were
suitable for PCA (p<0.001). Varimax with Kaiser normalization rotation was applied to maximize the
variances of the factor loadings across variances for each factor. The regression method was selected
for calculating the factor score coefficient. Three components with eigenvalue >1.0 were extracted. The
loadings were obtained from the eigenvalues of the three components and their corresponding
eigenvectors.





Because the samples collected in 1999 did not cover the whole traverse and the ion concentrations
were determined using a different IC system, the ion data of 1999 were excluded in the EF and PCA
analysis in the following.

**3 Results**
**3.1 Ion concentrations in surface snow**
Concentrations of ions in surface snow collected during the five seasons are shown in Fig. 2, and the
ranges (mean) of $Cl^-$, $NO_3^-$, $SO_4^{2-}$, $Na^+$, $NH_4^+$, $K^+$, $Mg^{2+}$ and $Ca^{2+}$ are 0.15-14.6 (1.29), 0.48-12.6 (3.37),
0.37-5.63 (1.52), 0.09-12.74 (0.68), 0.04-0.77 (0.16), 0.01-0.27 (0.04), 0.11-2.76 (0.22) and 0.01-0.50
(0.13) $\mu eq$ $L^{-1}$, respectively. These values fall within the reported ranges of the ITASE program
sampling (Bertler et al., 2005). Ion concentrations are both spatially and temporally variable, with the
coefficient of variation (ratio of one standard deviation over mean) of >0.48, suggesting a large
variability across the traverse. In general, ion concentrations do not follow a normal distribution
($p$>0.05, One-Sample Kolmogorov-Smirnov Test), with the values of skewness and kurtosis above 1.0,
but they correspond to a logarithmic normal distribution.
pH values of surface snow sampled in 2013 were measured with a glass pH electrode, and $H^+$
concentrations deduced from pH are correlated well with the values calculated from the ion-balance
method (Fig. 3(a)). On average, $H^+$ concentrations obtained from the ion balance approach are ~25 %
lower than those deduced from pH. It is noted that pH measurements in this study remain uncertain
considering that snow samples are highly undersaturated with respect to carbon dioxide ($CO_2$)
immediately after melting in the lab (Pasteris et al., 2012). On the other hand, organic acids, e.g.,
monocarboxylic and methanesulfonic acids (MSA), were excluded in $H^+$ calculation (Eq. 1), although
their concentrations in Antarctic snow tend to be very low (Li et al., 2015; Li et al., 2016). If the
contribution of organic acids to $H^+$ in the snow is negligible, the x-intercept of ~2.4 $\mu eq$ $L^{-1}$ in the linear
regression (Fig. 3(a)) can be regarded as the contribution from dissolved $CO_2$ in snow during pH
measurements. This value is close to that of pure water in equilibrium with $CO_2$ in the atmosphere,
with pH=5.6 corresponding to $H^+$ concentration of ~2.5 $\mu eq$ $L^{-1}$.
The percentage of each constituent to the total ions in surface snow is shown in Fig. 3(b). The most
abundant species is $H^+$, accounting for 39.6 % of the total ions, followed by $NO_3^-$ and $SO_4^{2-}$,
representing 27.5 and 12.5 % of the total ion budget, respectively. The high contribution percentage of
$H^+$ is consistent with previous investigations (Udisti et al., 2004; Traversi et al., 2009; Pasteris et al.,
2014), suggesting the acidic characteristics of surface snow. In general, ions $NH_4^+$, $K^+$, $Mg^{2+}$, and $Ca^{2+}$
are the smallest component of the ionic composition, with the four cation summing to ~5 % of the total.
Previous investigations of ions in surface snow covered various depths among different traverses or
campaigns, e.g., 1.0 m deep layer for the traverse from Terra Nova Bay to Dome C and top 25 cm snow
for the 1990 ITASE (Qin et al., 1992; Proposito et al., 2002). It is noted that different sampling depths
can result in varied ion concentrations in snow. For instance, in inland Antarctica, $NO_3^-$ is often
concentrated on the top few-centimeter snow, and decreases significantly with increasing depth (Shi et
al., 2015). Thus, any comparison of ion concentrations in surface snowpack collected from different
campaigns should be made with caution.

**3.2 Spatial patterns of ions in surface snow**
The spatial distribution patterns of ions on the traverse are consistent among the five campaigns (Fig.
2). In general, $Cl^-$, $Na^+$, $K^+$, and $Mg^{2+}$ show very high concentrations within the narrow coastal region,



and decrease sharply further inland, with low values on Dome A plateau (~1000-1250 km from the
coast). It is noted that some samples on the coast also show elevated $Ca^{2+}$ concentrations. The spatial
patterns are consistent with previous observations (Bertler et al., 2005; Kärkäs et al., 2005), and the
high ion concentrations near the coast have been explained by the strong marine air mass intrusions
(Hara et al., 2014).
Different from other species, $NO_3^-$ concentrations near the coast are low, and increase towards inland,
with the highest values on the Dome A plateau. A significant correlation is found between $NO_3^-$ and
distance from the coast, with $r = 0.56$ and $p < 0.001$. The spatial trend of $NO_3^-$ is generally opposite to
that of snow accumulation rate on the traverse (Figs. 2(a) and (c)), possibly associated with
post-depositional cycling of $NO_3^-$ in surface snow (Erbland et al., 2013; Shi et al., 2018b). Similarly,
there is a close relationship between $H^+$ and distance from the coast ($r = 0.48$, $p < 0.001$), suggesting a
higher acidity of inland snow. As for $SO_4^{2-}$, $NH_4^+$, and $Ca^{2+}$, no clear spatial trend was found on the
traverse.

**3.3 Ions in snow pits**
Clear seasonal cycles of $Na^+$ and $nssSO_4^{2-}$ are present in P1 and P2, and thus the two pits can be well
dated, spanning ~3 years (Figs. 4(a) and (b)). Based on the snow pit dating, it is estimated that snow
accumulation rate is ~50 (P1) and ~33 cm snow per year (P2), agreeing well with the field
measurements (P1: ~150 kg m$^{-2}$ a$^{-1}$; P2: ~100 kg m$^{-2}$ a$^{-1}$; Fig. 2(a)), assuming a snow density of ~0.33 g
cm$^{-3}$. At P1, negative $nssSO_4^{2-}$ values are observed in winter snow, i.e., $SO_4^{2-}/Na^+$ ratio below that of
bulk seawater, while all of the $nssSO_4^{2-}$ data in P2 pit are positive. It is difficult to assign the samples in
the snow pits to the four distinct seasons based on the measured parameters, and thus, in the following
discussion, we choose a conservative assignment method, i.e., a summer season featured with higher
$nssSO_4^{2-}$ and $SO_4^{2-}/Na^+$ ratio (and lower $Na^+$) and a winter season characterized with the opposite
patterns.
As for $nssSO_4^{2-}$ at P3, the very large signal at the depth of ~120 cm is most likely the fallout from the
massive eruption of Pinatubo in 1991 (Fig. 4(c)), based upon previous observations at Dome A (e.g.,
Hou et al., 2007). Accordingly, the snow accumulation rate from 1992 to 2010 is ~22 kg m$^{-2}$ a$^{-1}$, in line
with previous investigations (Hou et al., 2007; Jiang et al., 2012; Ding et al., 2016). Based on $nssSO_4^{2-}$
signals and the method proposed by Cole-Dai et al. (1997), 19 continuous samples have been identified
as influenced by Pinatubo eruption, covering ~2.5 years, possibly suggesting that the effects of
Pinatubo eruption on atmospheric chemistry lasted at least for 2.5 years over Dome A. Interestingly,
only elevated $SO_4^{2-}$ concentrations are present during this period, and anomalous high or low
concentrations of other ions are absent. Additionally, no correlation was found between $nssSO_4^{2-}$ and
other species during the 2.5 years, suggesting that Pinatubo volcanic emissions contribute less to the
ion budgets other than $SO_4^{2-}$ at Dome A.
Previous investigations proposed that $Na^+$ and $nssSO_4^{2-}$ in surface snow (top ~1 cm) collected during
a full year at central Antarctica show clear seasonal cycles, with high (low) $Na^+$ in winter (summer)
snow (Udisti et al., 2012). At P3, $Na^+$, $nssSO_4^{2-}$ and the ratios of $SO_4^{2-}/Na^+$ fluctuate significantly, and
these contrasts are unlikely indicative of the seasonal cycles as that for P1 and P2. In a full year of
snow accumulation at P3, on average, 7-8 samples were collected, allowing for examining the seasonal
variability of ions. Following the field measurements of snow accumulation rate at Dome A during
2008-2011 (~20 kg m$^{-2}$ a$^{-1}$; Ding et al., 2015), the snow samples covering the years 2008 and 2009 can
be roughly identified, assuming an even distribution of snow accumulation throughout the year. In total,



there are 7 and 8 samples identified in the years 2008 and 2009, respectively (Fig. 5), and no seasonal
cycles in $Na^+$, $nssSO_4^{2-}$, and $SO_4^{2-}/Na^+$ ratio were found, maybe related to the post-depositional
processes (e.g., migration, diffusion, and ventilation processes) and/or wind scouring that could
obscure the original signal (Cunningham and Waddington, 1993; Albert and Shultz, 2002; Libois et al.,
2014; Caiazzo et al., 2016).

**4. Discussions**
**4.1 Enrichment of ions in surface snow**

315       Statistics of enrichment factors (EFs) of ions in surface snow are shown in Fig. 6, and EFs ranges

(means) of $Cl^-$, $SO_4^{2-}$, $K^+$, $Mg^{2+}$, and $Ca^{2+}$ are 0.5-6.6 (1.8), 1.5- 87.8 (25.7), 0.7-11.4 (4.6), 0.9-6.2 (2.0),
and 0.2-63.2 (7.3), respectively. Most EFs of $Cl^-$, $K^+$, and $Mg^{2+}$ are close to 1.0, suggesting the main
source of sea salt spray, while most EFs of $SO_4^{2-}$ and $Ca^{2+}$ are well above 1.0, i.e., greatly enriched,
indicating additional sources. Spatially, EFs of $Cl^-$, $K^+$, $Mg^{2+}$, and $Ca^{2+}$ at the sites close to the coast are
around 1.0, with elevated values in interior areas, especially on the Dome A plateau.

321       Correlation plots of ions versus $Na^+$ in surface snow are shown in Fig. 7, and the plots above (below)

the seawater dilution line represent the enrichment (depletion) of the ions. The further the plots deviate
away from the line, the higher degree of enrichment or depletion of the ions. On the coast, most of the
$Cl^-/Na^+$ data are distributed close to the seawater dilution line (Fig. 7(a)), indicating a quantitative sea
salt tracer of snow $Cl^-$, while most of the plots in the interior areas are above the seawater line,
suggesting an enrichment of snow $Cl^-$. On this traverse, $nssCl^-$ accounted for an average of 38 % of
total $Cl^-$, with lower (higher) percentages on the coast (plateau), generally in line with previous reports
(e.g., Suzuki et al., 2002). The modifications in $Cl^-$ with respect to bulk seawater can occur via the
heterogeneous reactions, as follows (Finlayson-Pitts, 2003),
$NaCl + H_2SO_4 = HCl + Na_2SO_4$ (R1)
$NaCl + HNO_3 = HCl + NaNO_3$ (R2)
In the atmosphere, the production of HCl will result in depletion of $Cl^-$ in sea salt aerosol. The
'secondary' HCl, in the gas phase and/or fine aerosol mode, can be transported further inland due to the
longer lifetime (versus the coarse sea salt aerosols removed preferentially from the atmosphere). In this
case, an enrichment of $Cl^-$ would be expected in the inland snowpack. On the other hand, $Cl^-$ is not
irreversibly deposited to the snow, and it can be released back into the atmosphere through the
formation of HCl, resulting in an enrichment of $Cl^-$ in surface snow via re-deposition. Post-depositional
losses of HCl are thought to be associated with snow accumulation rate, with larger losses occurring at
sites with snow accumulation generally <40 kg $m^{-2}$ $a^{-1}$ (Röthlisberger et al., 2003). Indeed, a negative
correlation was found between snow accumulation and $nssCl^-$ (Fig. 8(a)) for most interior areas that
featured low snow accumulation and consequently an enhanced cycling of $Cl^-$.

342       Different from $Cl^-$, $Mg^{2+}$ is irreversibly deposited into the snow. Most of the $Mg^{2+}/Na^+$ data points

are above or close to the seawater dilution line, similar to that of $Cl^-/Na^+$ (Fig. 7(d)). On the coast,
$Mg^{2+}/Na^+$ data points are in general close to the seawater dilution line, suggesting the main source is
sea salt aerosols, while most of the inland samples are slightly enriched with $Mg^{2+}$, agreeing with
previous observations (e.g., Dome F; Hara et al., 2014). The fraction of $nssMg^{2+}$, on average,
represents ~36 % of $Mg^{2+}$ in snow, with lower (higher) values on the coast (plateau). The enrichment of
$Mg^{2+}$ has not been observed in sea salt particles produced by bubble bursting (Keene et al., 2007), and
thus enriched $Mg^{2+}$ in the snow is unlikely associated with sea salt spray. In the atmosphere, sea salt
aerosols would also be modified at low temperatures via the formation of mirabilite (R1), thus leading



to an elevated ratio of $Mg^{2+}/Na^+$ if mirabilite precipitate from the aerosols. However, the solid-liquid
separation of mirabilite in the aerosol droplet was not observed in the experiments (Wagenbach et al.,
1998a). Thus, the enrichment of $Mg^{2+}$ in surface snow is unlikely associated with sea salt fractionation.
Although it is proposed that $Mg^{2+}$ separation in sea salts can occur in surface snow due to the
re-freezing process on surface snow (i.e., the quasi-liquid layers on the crystal surface can act like
seawater freezing; Hara et al., 2014), our measurement of $Mg^{2+}$ in bulk snow is unlikely to support this
process responsible for $Mg^{2+}$ enrichment. A previous observation conducted near this traverse showed
a moderate correlation of $Mg^{2+}$ with element Al in the surface snowpack ($r$=0.53, p<0.05), indicating a
contribution of continental dust (Khodzher et al., 2014). Thus, the most plausible interpretation of
enriched $Mg^{2+}$ in surface snow is the contribution of terrestrial aerosols.
Similar to $Mg^{2+}$, most of $K^+/Na^+$ data points are close to the seawater dilution line on the coast,
suggesting a primary contribution of sea salt spray (Fig. 7(c)). Slightly enriched $K^+$ was present in
inland snow, possibly indicating other sources such as biological activity on the coast, mineral
transport, and combustion emissions in Southern Hemisphere (Rankin and Wolff, 2000; Virkkula et al.,
2006; Hara et al., 2013). Given that the sampling sites are at least several tens of kilometers away from
the coast, the contribution of biological activity to snow $K^+$ would be rather minor (Rankin and Wolff,
2000). A previous investigation of the atmospheric particles suggests a contribution of combustion in
South America and Southern Africa to atmospheric $K^+$ in Antarctica (Hara et al., 2013). Indeed, aerosol
particles from biomass burning in the Southern Hemisphere can be transported to Antarctica, resulting
in the ubiquitous distribution of biomass burning tracers observed in the snow on this traverse (Shi et
al., 2019). However, the average ratio of $nssK^+/nssCa^{2+}$ (~0.29) on the traverse is slightly higher than
that of the average crust (0.26; Bowen, 1979), likely supporting a minor contribution of biomass
burning emissions. If $nssK^+$ in surface snow is exclusively from terrestrial minerals and combustion
processes, it is estimated that ~10 % of $nssK^+$ is originated from biomass burning emissions.
$Ca^{2+}$ is generally enriched versus $Na^+$, with most of the $Ca^{2+}/Na^+$ data points above the seawater
dilution line, especially at inland sites (Fig. 7(e)). The fraction of $nssCa^{2+}$, on average, accounts for
~77 % of total $Ca^{2+}$ in surface snow, indicating other dominant sources. In Antarctica, snow $nssCa^{2+}$
has been thought to be mainly associated with terrestrial inputs (Bertler et al., 2005; Wolff et al., 2010).
Previous modeling studies suggest that the dust mass reaching East Antarctica mainly originates from
South America, specifically Patagonia (Basile et al., 1997; Wolff et al., 2006; Mahalinganathan and
Thamban, 2016). Metal isotopes in snow collected on this traverse suggested that Australian mineral
dust also can contribute to snow particles (Du et al., 2018). In addition, Antarctic ice free areas were
thought to be a contribution to snow dust (Delmonte et al., 2013; Du et al., 2018). If the dust mass
originated from ice free area near the coast and dominated $nssCa^{2+}$, then $nssCa^{2+}$ concentrations near
the coast would be expected to be higher, while the data shows the opposite. Thus, terrestrial dust mass,
possibly from both South America and Australia likely dominates snow $nssCa^{2+}$.
$SO_4^{2-}$ is greatly enriched in all surface snow (Fig. 7(b)), together with the minimum sea ice coverage
around East Antarctica in late summer (Holland et al., 2014), suggesting that sea salts in surface snow
are from open seawater rather than from the sea ice. On the traverse, $nssSO_4^{2-}$ represents 33-99 %
(mean=95 %) of total $SO_4^{2-}$ in surface snow, with lower (higher) proportions on the coast (plateau). In
Antarctica, $nssSO_4^{2-}$ essentially originates from marine biogenic production of DMS (Saltzman, 1995)
and occasionally from explosive volcanism (Cole-Dai et al., 2000; Cole-Dai et al., 2013). In this study,
the significant enrichment of $SO_4^{2-}$ suggests a dominant role of ocean bioactivities. Different from the
coarse sea salt aerosols, $nssSO_4^{2-}$ can form fine aerosol particles in the atmosphere (Legrand et al.,



2017a), resulting in long atmospheric residence time (>10 days to weeks) and consequently efficient
transport (Bondietti and Papastefanou, 1993; Hara et al., 2014). This can help explain the elevated
deposition flux of $nssSO_4^{2-}$ frequently found at inland Antarctic sites, e.g., site P2 (discussed below).
On this transect, a negative relationship was found between snow accumulation rate and $SO_4^{2-}$ (or
$nssSO_4^{2-}$) (Figs. 8(c) and (d)), suggesting that snow accumulation rate can influence snow $SO_4^{2-}$
concentration, possibly via dilution effects, but overall <~10 % of the variation in $SO_4^{2-}$ concentrations
can be explained by the relationship.
The ternary diagram of $Cl^-$, $Na^+$, and $SO_4^{2-}$ can well characterize the modification processes to sea
salt aerosols, and the ternary plot of the three ions in surface snow is shown in Fig. 9. The values of the
ions were normalized via the following equation,
$X=[X]/([Na^+]+[Cl^-]+[SO_4^{2-}])$ Eq. (4),
where [X] is the concentration of ion X in the snow (in $\mu eq\ L^{-1}$). The dashed line between the seawater
reference value and the $SO_4^{2-}$ vertex represents the sea salt aerosol composition with additional $SO_4^{2-}$,
i.e., the ratio of $Cl^-/Na^+$ keeps constant (1.17) with additional $SO_4^{2-}$ along the dashed line. The presence
of acids ($HNO_3$ and $H_2SO_4$) would result in the liberation of HCl into the atmosphere via reactions R1
and R2, resulting in the changes in Cl/Na ratios, i.e., either $Cl^-$ loss or gain are located right or left of
the line, respectively. It is shown that all of the data points are above the seawater plot, suggesting an
enrichment of $SO_4^{2-}$ in surface snow. Most of the data points are located left of the line, indicating the
general enrichment of $Cl^-$ due to reactions R1 and R2 occurring in the atmosphere and/or in the
snowpack. But the coastal data points are generally close to the line, suggesting that the degree of sea
salt modification is generally low in the snow.

**4.2 Groups of ions in surface snow**
PCA is a powerful tool for identifying the common sources and/or transport process of chemicals in
different environments. The PCA results (i.e., loadings in each PC), communalities, initial eigenvalues,
and explained cumulative percent of the ions in surface snow are listed in Table 1. The first three PCs
accounted for 76 % of the variation of the eight original variables. PC1 accounts for 46 % of the
variance and is highly loaded by $Cl^-$, $Na^+$, $K^+$, and $Mg^{2+}$, with the factor loadings higher than 0.7. In
addition, the four species are correlated well with each other (Table 2), suggesting the variation of the
four species is dominated by sea salt aerosols, consistent with the EFs results. Thus, PC1 is indicative
of the origin of sea salt aerosols.
PC2 accounts for 17 % of the total variance, and the loading values of $NH_4^+$ and $Ca^{2+}$ in PC2 are high,
~0.8. In Antarctic snow, $NH_4^+$ is thought to be mainly associated with biological decomposition of
organic matter in the Southern Ocean (Johnson et al., 2007; Kaufmann et al., 2010). In addition,
biomass burning from mid-latitudes can contribute to snow $NH_4^+$ in West Antarctica (Pasteris et al.,
2014). On this transect, no correlation was found between $NH_4^+$ and biomass burning tracers on the
traverse (Shi et al., 2019), suggesting a minor role of biomass burning emissions. Thus, $NH_4^+$ in surface
snow tends to be dominated by marine biological activities, and elevated $NH_4^+$ concentrations in
summer snow would be expected, e.g., summer mean of 0.23 $\mu eq\ L^{-1}$ versus winter mean of 0.16 $\mu eq$
$L^{-1}$ at P1 (Fig. S1). It is proposed that the transport of $NH_4^+$ via free troposphere is an important
pathway (Kaufmann et al., 2010). Similarly, the meridional transport of particle mass from continents
to Antarctica is more efficient in the mid-troposphere (Krinner and Genthon, 2003; Krinner et al., 2010;
Shi et al., 2019). Thus, the shared transport process may explain, at least in part, the positive loadings
of $NH_4^+$ and $Ca^{2+}$ in PC2.

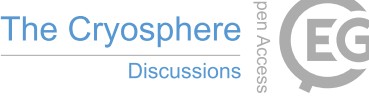

$NO_3^-$ is highly loaded in PC3, which accounts for 13 % of the system variance. On this traverse, $NO_3^-$
in the snow has been extensively investigated, and it is proposed that $NO_3^-$ concentrations were
influenced by post-depositional processing which is largely dependent on snow accumulation rate (Shi
et al., 2015; Shi et al., 2018a; Shi et al., 2018b). A negative relationship was found between $NO_3^-$ and
snow accumulation rate (Fig. 8(b)), suggesting a high degree of $NO_3^-$ cycling driven by photolysis at
low snow accumulation sites.
$SO_4^{2-}$ did not show high loadings in any of the three extracted components. Its positive loading in
PC1 (0.55) and weak relationships between $SO_4^{2-}$ and sea salts ($Cl^-$ and $Na^+$) likely supports the
contribution of sea salt aerosols, although a minor one. A positive loading of $SO_4^{2-}$ is also present in
PC3 (0.42), and a weak correlation was found between $SO_4^{2-}$ and $NO_3^-$. In summer, snow $NO_3^-$ is
mainly produced locally considering that the long-range transported nitrogen compounds can
decompose and undergo rapid $NO_x$ cycling in the local boundary layer, and $NO_3^-$ production is closely
related to the atmospheric oxidants (Davis et al., 2004; Jones et al., 2011; Morin et al., 2011; Björkman
et al., 2014). Although the main production pathways of $NO_3^-$ and $SO_4^{2-}$ are different from each other
(Ishino et al., 2017), their formations are closely related to certain oxidant abundances in the
atmosphere (e.g., OH radical), which may partly account for the positive loading of $SO_4^{2-}$ in PC3.

**4.3 Ion fluxes and enrichment in snow pits**
In this section, we discuss the fluxes and enrichment of ions at different depths (i.e., summer and
winter snow) in the three snow pits. The bottom ~30 cm layer of P3 will be excluded in the discussion,
since it represents a snow layer clearly impacted by volcanic (Pinatubo) eruption emissions.
Ion fluxes in snow can be determined by multiplying the concentrations by snow accumulation rate,
and the results in the 3 snow pits are shown in Fig. 10. The highest fluxes of ions except for $NO_3^-$ were
present at P1, followed by P2 and P3. The flux of $NO_3^-$ shows a different pattern, with the highest value
at P2, possibly due to the redistribution of $NO_3^-$ across the Antarctic ice sheet driven by photolysis (Shi
et al., 2018b). It is noted that $nssSO_4^{2-}$ fluxes at P1 (99.4±46.7 µeq m$^{-2}$ a$^{-1}$) and P2 (109.2±21.6 µeq m$^{-2}$
a$^{-1}$) are comparable, although P1 is located on the coast and P2 located further inland (~800 km from
the coast). In addition, the ratio of $nssSO_4^{2-}$ flux at P1 over that at P3 is 2.2, the lowest value among the
ratios for the observed ions (17.2, 7.5, 26.7, 8.5, 17.4, 17.0, and 10.0 for $Cl^-$, $NO_3^-$, $Na^+$, $NH_4^+$, $K^+$,
$Mg^{2+}$, and $Ca^{2+}$, respectively), suggesting more efficient transport of $nssSO_4^{2-}$. In other words,
atmospheric $nssSO_4^{2-}$ from the open ocean can be efficiently transported to at least as far inland as
~800 km from the coast (~2800 m above sea level; site P2).
At P1, the plots of $Cl^-$, $K^+$, $Mg^{2+}$, and $Ca^{2+}$ versus $Na^+$ are all close to the bulk seawater dilution line
(Fig. 11), with EFs of the four species generally below 3. In addition, the slope values of the linear
regression between $Na^+$ and the four ions are close to those of seawater, suggesting a dominant source
of sea salt aerosols. As for $SO_4^{2-}$ in the snow, the proportion of $nssSO_4^{2-}$ to $SO_4^{2-}$ is much higher in
summer (~86 %) than in winter (~27 %). All $nssSO_4^{2-}$ in summer snow is positive, while some winter
snow samples featured negative $nssSO_4^{2-}$, i.e., $SO_4^{2-}/Na^+$ ratio below the value of seawater (Fig. 4(a)),
suggesting sea salt aerosols in winter from sea ice (Marion et al., 1999). In the winter snow, if all of the
$SO_4^{2-}$ is from sea salt aerosols, $nssSO_4^{2-}$ is expected to be lower than or close to zero. However, 13 out
of the 17 samples classified as winter snow at P1 were characterized with positive $nssSO_4^{2-}$, suggesting
a significant contribution from marine biogenic emissions. It is interesting that $nssSO_4^{2-}$ has a strong
negative correlation with $Na^+$ in winter snow (r=0.82, p<0.001), raising two potential cases: 1) stronger
winds transport more sea salt aerosols to P1 featured with depleted $SO_4^{2-}$ from sea ice, thereby resulting



in low concentrations of $nssSO_4^{2-}$ and assuming a stable $SO_4^{2-}$ input flux from marine biogenic
emissions; and/or 2) with a larger extent of sea ice and strong transport, a large sea salt flux would still
result but carry less $nssSO_4^{2-}$ from marine biogenic emissions due to the longer transport distance
(Wolff et al., 2006 and references therein). If case 2) dominated $nssSO_4^{2-}$ variations in the winter snow,
lower $nssSO_4^{2-}$ would be expected in the end than at the beginning of winter when a sea ice coverage
minimum is present. The observation at P1, however, does not support this expected season trend (Fig.
S2). It is most likely, then, that sea salt aerosol inputs dominate $nssSO_4^{2-}$ variations in the winter snow
instead of the marine biogenic emissions.
The patterns of relationships between ions and $Na^+$ at P2 are similar to those of P1 except for $Ca^{2+}$
(Fig. 11). EFs of $Cl^-$, $K^+$, and $Mg^{2+}$ at P2 are 1.5±0.2, 1.4±0.5, and 1.5±0.3 (mean±1σ), respectively,
suggesting again a main source of sea salt aerosols. EFs of $Cl^-$ are slightly higher in summer snow
(1.57) than in winter snow (1.47), possibly indicating the presence of elevated $H_2SO_4$ and $HNO_3$ during
summer promoting the production of HCl via R1 and R2 (discussed above). $Ca^{2+}$ is enriched in P2
(EFs=6.1±2.8), and it remains relatively constant with increasing $Na^+$ (Fig. 11), possibly suggesting
seasonal variations in terrestrial dust inputs are insignificant. As for $SO_4^{2-}$, it is significantly enriched,
with the EFs of 18.6±11.4, and the fractions of $nssSO_4^{2-}$ to $SO_4^{2-}$ in summer and winter snow are 95
and 89 %, respectively. The very high $SO_4^{2-}$ to $Na^+$ ratio in winter (~1.6, versus 0.12 of bulk seawater)
suggests that marine biogenic emissions dominate $SO_4^{2-}$ other than the sea salt aerosols, different from
that at P1. It is suggested that the sea salt aerosol flux from the sea ice in winter is much lower in the
inland Antarctica than on the coast. Previous investigations proposed that sea salt aerosols emitted from
sea ice are an important contribution to sea salt budget in central Antarctica in winter (Levine et al.,
2014; Legrand et al., 2016; Legrand et al., 2017b). Here, our data indicate that marine emissions could
also be an important source.
At P3, $Cl^-$, $K^+$, and $Mg^{2+}$ are correlated well with $Na^+$ (Fig. 11), and EFs of the 3 ions are 2.1±1.0,
2.6±1.3, and 2.0±0.8, respectively, higher than those of P2. Although the sea salt fractions of $Cl^-$, $K^+$,
and $Mg^{2+}$ account for most of their total budgets in the snow, the other sources can occasionally be
important. On average, $nssCl^-$ accounts for ~40 % of the total $Cl^-$, suggesting that, $Cl^-$ at Dome A is
mainly from the sea salt aerosols, but the deposition of HCl is also an important contribution. This
percentage is higher than that at P2 (~30 %), suggesting a more important role of HCl on $Cl^-$ budget in
further inland snow. $Ca^{2+}$ is enriched noticeably at P3, with EEs of 6.6±5.0, close to that of P2,
suggesting the terrestrial particle mass as the primary source. In terms of $SO_4^{2-}$, it is enriched
significantly (EFs of 27.4±17.3), and the non-sea salt fraction accounts for ~95 % of total $SO_4^{2-}$,
comparable to that of P2. At P2 and P3, the negative $nssSO_4^{2-}$ signal resulted from sea salt aerosols
originated from sea ice has been completely swamped by the biogenic $SO_4^{2-}$, generally in line with the
observation at Dome C (Udisti et al., 2012).
The ternary plots of $Cl^-$, $Na^+$, and $SO_4^{2-}$ at the three pits are shown in Fig. 12. At P1, all plots are
close to the seawater composition line, suggesting the modification processes to sea salt aerosols is
negligible, similar to that of coastal surface snow. Several winter snow samples at P1 show a depletion
of $SO_4^{2-}$ relative to seawater, associated with the precipitation of mirabilite during sea ice formation,
while more additional $SO_4^{2-}$ is present in summer snow (Fig. 12(a)). In general, patterns of the three
ions at P2 are similar to those of P1, but with $Cl^-$ enriched, especially in summer snow (Fig. 12(b)).
Similarly, enriched $Cl^-$ was observed at P3 (Fig. 12(c)), associated with scavenging of HCl in the
atmosphere by snow. Such a pattern implies the ubiquitous modification process to sea salts in inland
Antarctica throughout the year (via R1 and R2). Together with the surface snow observations (Fig. 9),



$Cl^-$ in the interior areas, often deviating from the seawater dilution line remarkably, is not a quantitative
indicator of sea salts in snow. At P3, the data points are closer to the $SO_4^{2-}$ summit in comparison with
the other two sites, possibly suggesting predominant $H_2SO_4$ scavenging (e.g., Mahalinganathan et al.,
530    2011).

**4.4 Interannual variations of ions in surface snow**

As the snow sampling protocols (e.g., sampling snow depth and intervals) on the traverse are the
same in different years, we can directly compare ion concentrations in surface snow collected during
different campaigns. Independent samples t test showed that concentrations of $Cl^-$, $Na^+$, $K^+$, $Mg^{2+}$, and
$SO_4^{2-}$ in surface snow are generally higher in 2015 than in the other years, while $Ca^{2+}$ exhibited an
opposite pattern ($p < \sim 0.05$; Table S1). (2019 data are from personal communication with S. Lu, 2020)
Averaged ion concentrations in surface snow collected in different campaigns are shown in Fig. 13.
As for the sea salt related ions in surface snow, i.e., $Cl^-$, $Na^+$, $K^+$, and $Mg^{2+}$, their concentrations are
largely dependent on the transport strength of sea salt aerosols from the oceans, which is strongly
linked to large scale atmospheric and oceanic circulation at the high southern latitudes (Goodwin et al.,
2004; Russell and McGregor, 2010). We hypothesize that snow sea salt concentrations in Princess
Elizabeth Land (PEL), where most of the investigation sites in this study are located (Fig. 1), are
related to the variations in the sea level pressure over the Southern Indian Ocean, specifically the
Southern Indian Ocean low (SIOL), a quasi-stationary climatological feature located north of Prydz
Bay (Xiao et al., 2004). To test this hypothesis, interannual variation in the SIOL during the study
period was examined with the aid of the ERA-interim reanalysis. To quantify the strength of the SIOL,
the circulation indices of a closed pressure system, including the area index (S) and strength index (P),
were calculated following Wang et al. (2007). Considering that the sampling time was January and
February, the austral summertime mean sea level pressure was used to calculate the circulation indices
in each year (Fig. S3). It is shown that the SIOL is stronger in the austral summer of 2014/2015, i.e.,
the larger area and the greater strength of SIOL (Fig. S3(f)). A significant correlation was found
between the area index of SIOL (S) and $Na^+$ ($r=0.89$, $p=0.03$; Fig. 13(a)). Accordingly, the higher
concentrations of sea salts observed in 2015 can, at least in part, be explained by the SIOL anomaly.
Indeed, the marine air mass intrusion into the continent is associated with large scale boundary-layer
turbulence over the ocean or blocking anticyclones (Naithani et al., 2002; Goodwin et al., 2004), and
higher snow sea salt concentrations in coastal PEL were generally connected to the deepening of SIOL
(Xiao et al., 2004).
Similar to the temporal patterns of sea salts, higher $SO_4^{2-}$ (and $nssSO_4^{2-}$) concentrations were also
observed in 2015. This is likely associated with the fact that $SO_4^{2-}$ in surface snow is mainly from
marine biogenic emissions, and a stronger SIOL would also promote the transport of $SO_4^{2-}$.
Interestingly, $Ca^{2+}$ exhibits the lowest concentration in 2015, which may be related to that $Ca^{2+}$ is
mainly originated from the mid-latitude terrestrial particle mass, instead of the Southern Ocean
emissions. A stronger polar low (e.g., SIOL) usually corresponds to strengthening westerly winds, and
thereby would result in weaker meridional transport from the mid-latitudes (Marshall, 2003; Goodwin
et al., 2004; Jones et al., 2009).
It is noted that the observation covers a relatively short period of time, and changes in snow
accumulation rate and transport strength from year to year are also likely to influence the variability of
ions on an interannual timescale. Thus, there is still uncertainty on the relationships between ion
concentrations and the SIOL. During the observation period (2011-2019), however, the SIOL is likely



an important factor influencing the interannual variability of major ions in surface snow on the traverse.

**5 Conclusions**

Surface snow and snow pit samples collected on a traverse from coastal Zhongshan Station to the ice sheet summit, East Antarctica, during five campaigns were used to comprehensively investigate spatial and temporal variations in snow chemistry. It is shown that $Cl^-$, $Na^+$, $K^+$, and $Mg^{2+}$ concentrations are high within the narrow coastal region, falling off strongly further inland, while $NO_3^-$ exhibits an opposite trend and no clear spatial trends were found for $SO_4^{2-}$, $NH_4^+$, and $Ca^{2+}$. In inland snow, $Cl^-$, $K^+$, and $Mg^{2+}$ are slightly enriched relative to $Na^+$ with respect to the composition of seawater. The enrichment of $Cl^-$ is likely associated with the deposition of HCl produced from dechlorination of sea salt aerosols, and enriched $K^+$ and $Mg^{2+}$ are possibly linked to terrestrial particle mass. $Ca^{2+}$ and $SO_4^{2-}$ are significantly enriched versus $Na^+$, and terrestrial dust mass and marine biogenic emissions are responsible for the enrichments respectively. Snow $NH_4^+$ is related to marine biological activities, and multivariate statistical analysis suggests, at least in part, the $NH_4^+$ transport is via free troposphere.

In coastal snow pit, parts of the winter snow showed a depletion of $SO_4^{2-}$ versus $Na^+$, indicating sea salt aerosols sourced from sea ice. In the interior areas, although sea salt aerosols originated from sea ice contribute to a significant depletion of $SO_4^{2-}$, the negative $nssSO_4^{2-}$ signal has been completely swamped by the contribution from biogenic $SO_4^{2-}$. In addition, $Cl^-$ in the snow is more enriched in summer than in winter, possibly related to more HCl formation due to elevated acid concentrations during summertime. Ternary plots of $Cl^-$, $Na^+$ and $SO_4^{2-}$ in snow suggest the modification process to sea salts is negligible on the coast, while the degree of modification to sea salts is higher in inland throughout the year, which results in $Cl^-$ not being a quantitative indicator of sea salts. Ion flux assessment suggests an efficient transport of $nssSO_4^{2-}$ to at least as far inland as the ~2800 m contour line. With the aid of reanalysis, it is found that the interannual variations in ion concentrations in surface snow are likely connected to changes in the Southern Indian Ocean low from year to year.

**Data availability**. This dataset, chemical data on ion concentrations in snow on the traverse from coast (Zhongshan Station) to Dome A, is in the process of being hosted on a public server by the Chinese National Arctic and Antarctic Data Center (https://www.chinare.org.cn/).

**Author contributions.** GS, ZC, YL and BS designed the experiments and GS, HM, ZH, CA, SJ, TM, JY, DW and SL carried them out. GS and MH prepared the manuscript with contributions from all co-authors.

**Competing interests.** The authors declare that they have no conflict of interest.

**Acknowledgements**

This research was supported by the National Science Foundation of China (Grant Nos. 41922046 and 41576190 to GS; Grant No. 41876225 to HM) and the National Key Research and Development Program of China (Grant No. 2016YFA0302204 to GS). The authors are grateful to the CHINARE inland members for logistic support and assistance.



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



**Table 1** Rotated component matrix of the major ions in surface snow. (Extraction method: principal
component analysis. Rotation method: varimax with Kaiser normalization. Rotation converged in 4
iterations.) Factor loadings were calculated from the eigenvalues of the three components and their
corresponding eigenvectors, and the values greater than 0.7 are shaded.

| Chemical ions | PC1 | PC2 | PC3 | Communalities |
|---|---|---|---|---|
| $Cl^-$ | 0.93 | -0.01 | 0.27 | 0.93 |
| $NO_3^-$ | 0.04 | -0.06 | 0.95 | 0.90 |
| $SO_4^{2-}$ | 0.55 | 0.08 | 0.42 | 0.49 |
| $Na^+$ | 0.98 | -0.01 | -0.06 | 0.96 |
| $NH_4^+$ | 0.10 | 0.81 | 0.04 | 0.66 |
| $K^+$ | 0.71 | 0.25 | 0.12 | 0.57 |
| $Mg^{2+}$ | 0.96 | 0.05 | -0.09 | 0.92 |
| $Ca^{2+}$ | 0.03 | 0.79 | -0.07 | 0.62 |
| Initial eigenvalues | 3.67 | 1.33 | 1.06 | |
| Percentage of variance | 46 | 17 | 13 | |
| Cumulative percent | 46 | 63 | 76 | |







**Table 2** Pearson correlation matrix of major ions in surface snow

| | $Cl^-$ | $NO_3^-$ | $SO_4^{2-}$ | $Na^+$ | $NH_4^+$ | $K^+$ | $Mg^{2+}$ | $Ca^{2+}$ |
|---|---|---|---|---|---|---|---|---|
| $Cl^-$ | 1.00 | 0.24[**] | 0.47[**] | 0.94[**] | 0.05 | 0.74[**] | 0.91[**] | 0.09 |
| $NO_3^-$ | | 1.00 | 0.21[**] | -0.02 | -0.04 | 0.09[*] | -0.04 | -0.05 |
| $SO_4^{2-}$ | | | 1.00 | 0.34[**] | 0.08 | 0.30[**] | 0.31[**] | 0.03 |
| $Na^+$ | | | | 1.00 | 0.05 | 0.77[**] | 0.98[**] | 0.12[*] |
| $NH_4^+$ | | | | | 1.00 | 0.19[**] | 0.10[*] | 0.30[**] |
| $K^+$ | | | | | | 1.00 | 0.75[**] | 0.15[**] |
| $Mg^{2+}$ | | | | | | | 1.00 | 0.15[**] |
| $Ca^{2+}$ | | | | | | | | 1.00 |

[**]. Correlation is significant at the 0.01 level (2-tailed).
[*]. Correlation is significant at the 0.05 level (2-tailed).







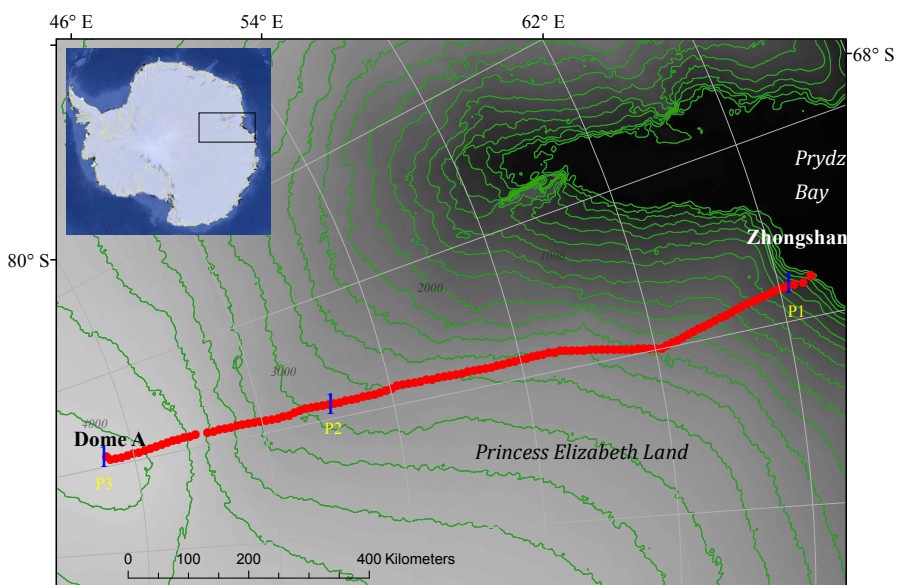

**Figure 1.** The Chinese inland investigation traverse from the coast (Zhongshan station) to the ice sheet
summit, Dome A, East Antarctica. The traverse is generally along the 77.0 °E longitude.





**Figure 2.** Annual snow accumulation rate, elevation (a) and ion concentrations in surface snow
collected during five seasons (b-i). Annual snow accumulation rate is obtained from field bamboo stick
measurements, updated to 2016 from Ding et al. (2011). The closed diamond, open circle, closed
triangle, cross and closed circle denote ion concentrations in the years 1999, 2011, 2013, 2015, and
2016, respectively. Note that a base-10 log scale is used for the $y$-axis of $Cl^-$ (b), $Na^+$ (e), and $Mg^{2+}$ (h).






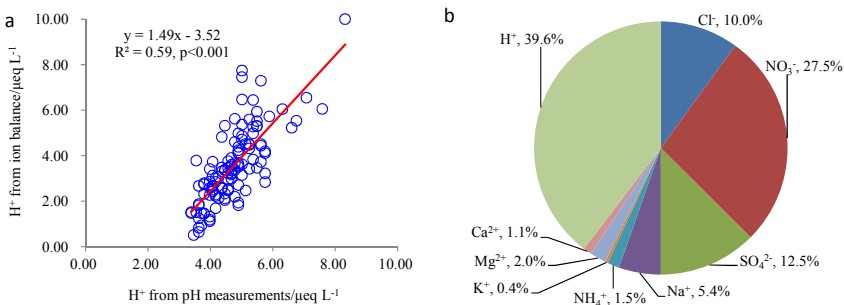


**Figure 3.** Major ions in surface snow on the Chinese inland Antarctic traverse. Concentrations of $H^+$
derived from pH versus those from the ion balance method are shown in panel (a), and contribution
percentages of each ion to the total are shown in panel (b), in $\mu eq\ L^{-1}$.












**Figure 4.** Profiles of $SO_4^{2-}$, $Na^+$, and $SO_4^{2-}/Na^+$ ratios in snow pits P1 (a), P2 (b), and P3 (c). Red and
blue arrows in panels (a) and (b) represent the middle of the identified summer and winter seasons,
respectively, and shaded areas denote summer seasons (see text). The red dashed line in panel (a)
represents the ratio of $SO_4^{2-}/Na^+$ in bulk seawater, while the red dashed line in panel (c) signifies the
first snow sample significantly influenced by the Pinatubo eruption. One seasonal cycle generally
represents local $Na^+$ minima and $nssSO_4^{2-}$ and $SO_4^{2-}/Na^+$ maxima.






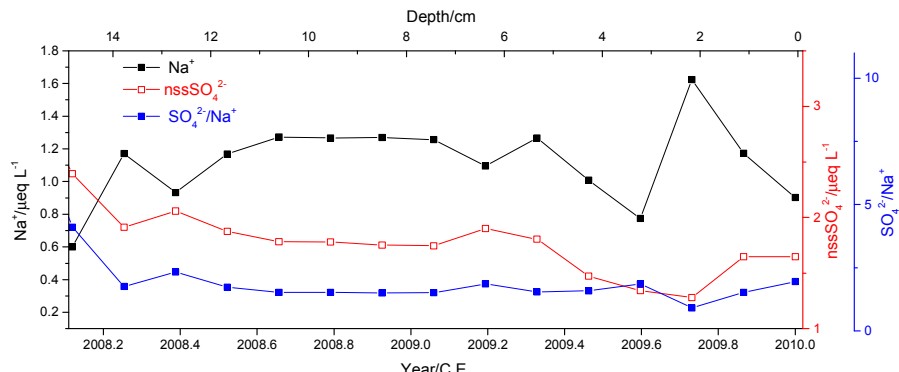


**Figure 5.** Variations in $Na^+$, $nssSO_4^{2-}$, and $SO_4^{2-}/Na^+$ ratio in the snow in the years 2008 and 2009, in Dome A snow pit (P3).









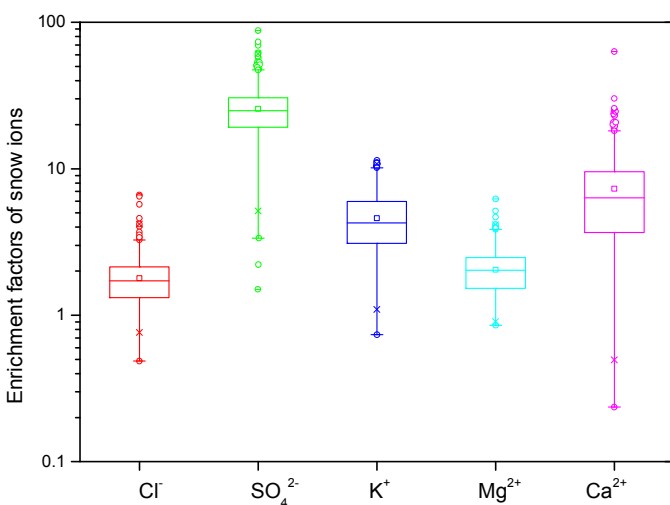


**Figure 6.** Statistics of enrichment factors of ions in surface snow. Box and whisker plots represent
maximum (top end dash symbol for each box), minimum (bottom end dash symbol for each box), the
range 1-99 % (top and bottom X symbol for each box), percentiles (5th, 25th, 75th, and 95th), and
median (50th, solid line) and mean (open square near the center of each box). Note that the data outside
the range 5-95 % are shown as open circles.



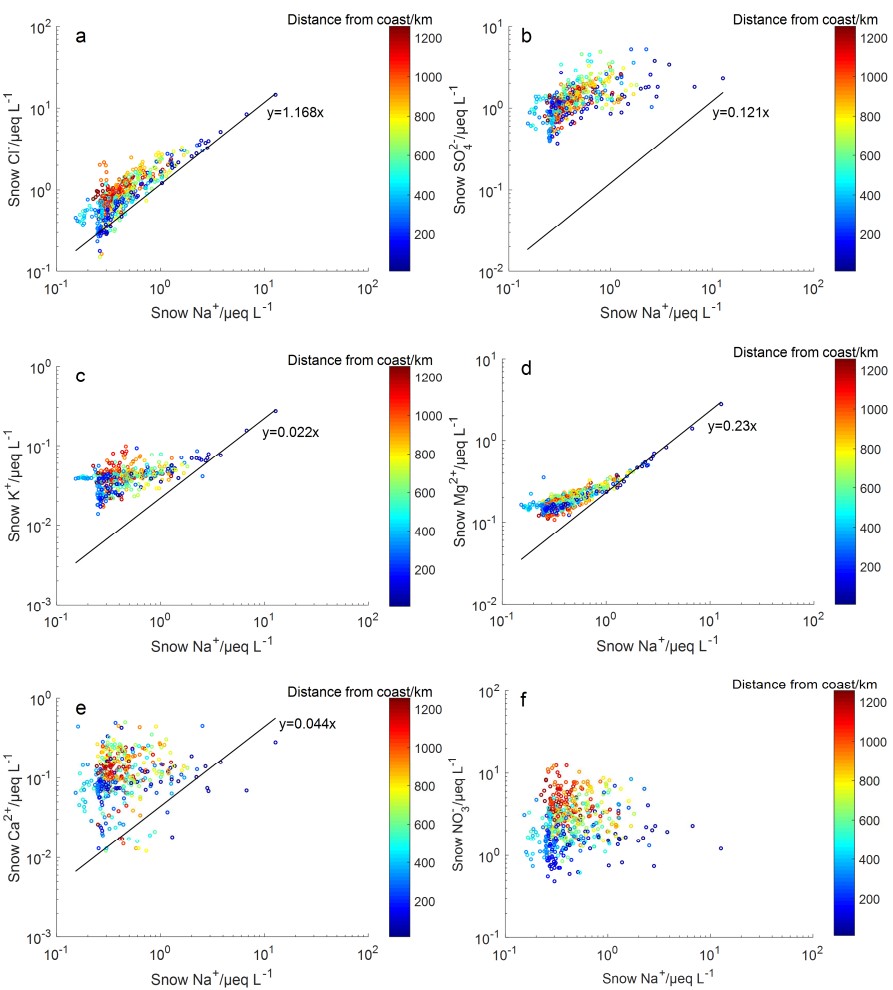

927

**Figure 7.** Correlation plots of Cl⁻, SO₄²⁻, K⁺, Mg²⁺, Ca²⁺, and NO₃⁻ versus Na⁺ in surface snow. The black solid line represents the seawater dilution line, with slopes of typical ions versus Na⁺ ratios in seawater (in µeq L⁻¹). The concentration of NO₃⁻ in seawater is too variable among the seas, and a representative ratio of NO₃⁻/Na⁺ cannot be presented. Note that a base-10 log scale is used for ion concentrations.







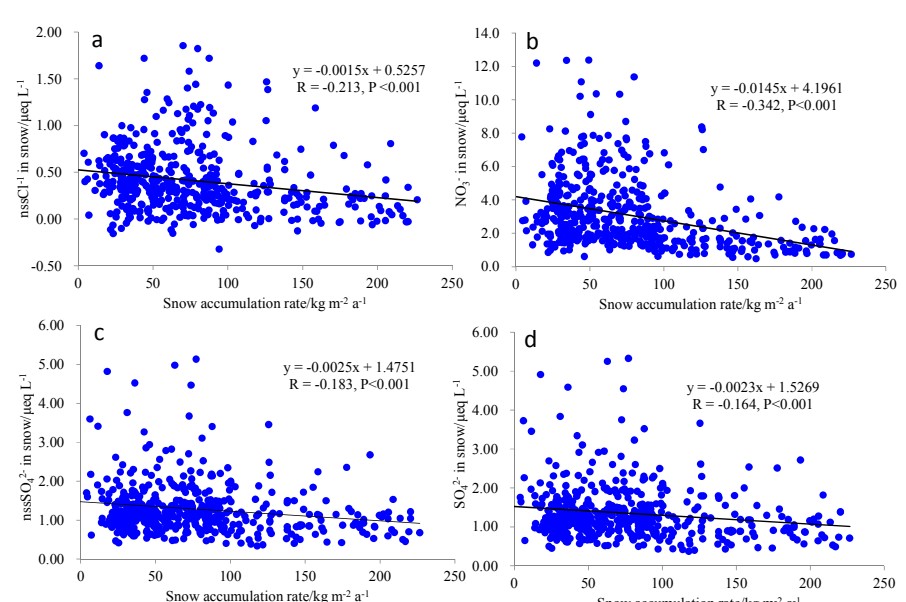


**Figure 8.** Relationship between chemical ions in surface snow and snow accumulation rate on the
traverse.








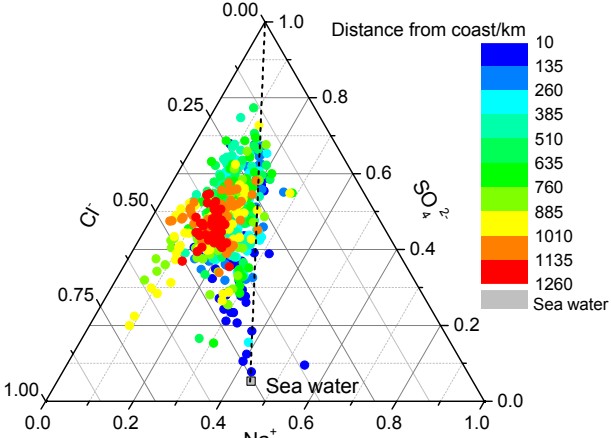


**Figure 9.** Ternary plot of $Cl^-$, $Na^+$, and $SO_4^{2-}$ in surface snow samples. Bulk seawater composition is denoted by a grey square. The dashed line extending between the sea salt reference value and the $SO_4^{2-}$ summit represents the composition of sea salt with increasing $SO_4^{2-}$.







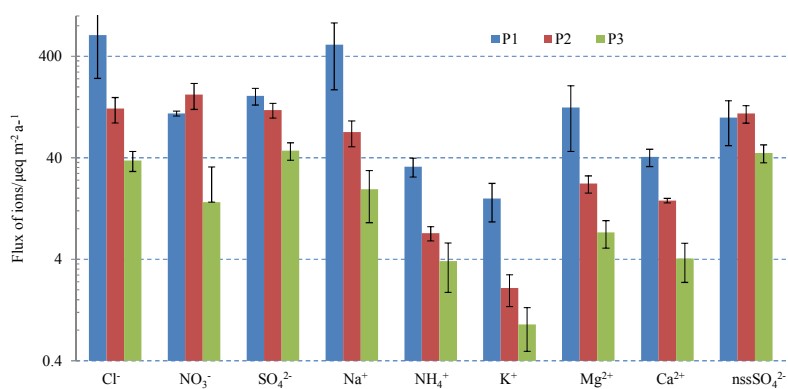


**Figure 10.** Ion fluxes at the three pits (P1, P2, and P3). The error bars represent one standard deviation
of fluxes in different years. Note that a base-10 log scale is used for the *y*-axis.





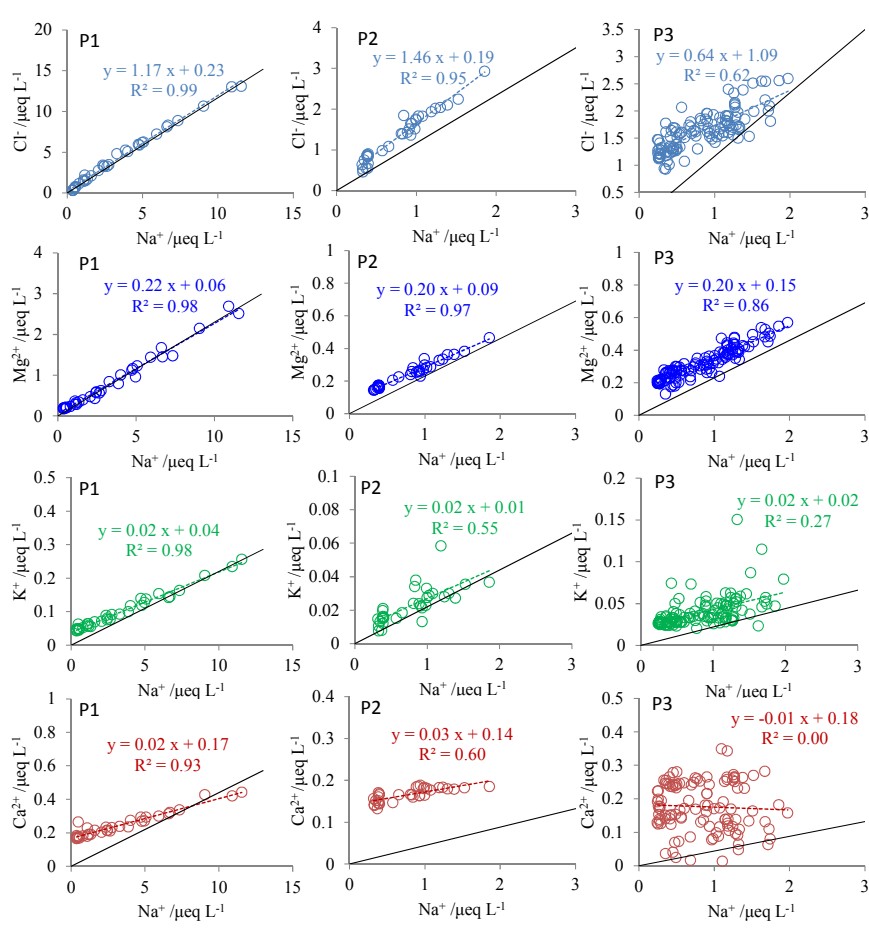


**Figure 11.** Relationships between $Na^+$ and $Cl^-$, $K^+$, $Mg^{2+}$, $Ca^{2+}$ in the three snow pits (P1, P2, and P3). Also shown are the linear regressions between them (dashed line), with all of the linear correlation significant at $p<0.001$ except $Ca^{2+}/Na^+$ at P3. The black solid line represents seawater dilution line.







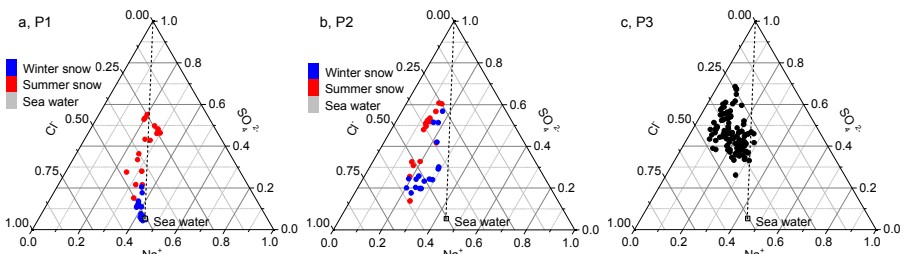

**Figure 12.** The same as Fig. 7, with blue and red dots in panels (a) and (b) representing winter and summer snow, respectively.






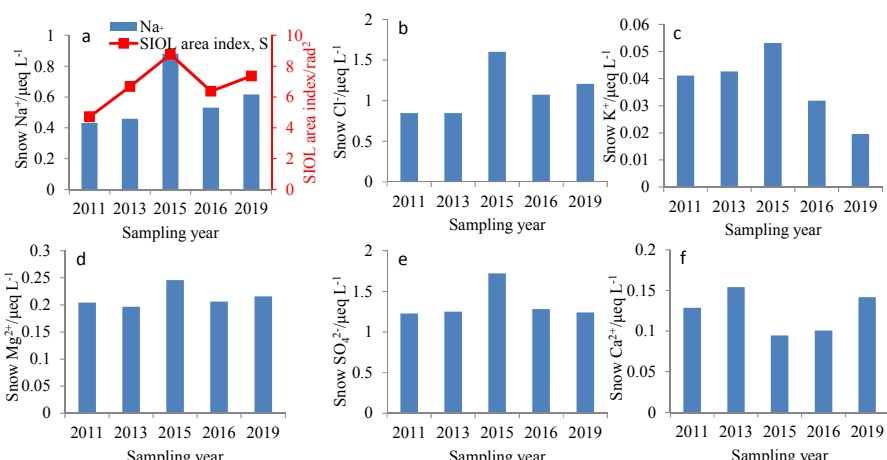


**Figure 13.** Averaged ion concentrations in surface snow collected on the traverse in different years.
The area index of the Southern Indian Ocean low (SIOL), S, is shown in panel (a), calculated following
Wang et al. (2007). The mean sea level pressure from ERA-interim reanalysis during the austral
summers in 2010/2011, 2012/2013, 2014/2015, 2015/2016, and 2018/2019 was used to calculate the
values of S (Figure S3).