# Peer review of "Spatial and temporal variations in snow chemistry along a traverse from coastal East Antarctica to the ice sheet summit (Dome A)"

_The Cryosphere, 2020_

## Referee Comment (RC1) · Anonymous Referee #1 · 27 Oct 2020

This paper uses surface samples from a series of traverses between the coast and the inland plateau Dome A station, along with 3 pit sequences, to study the geographic pattern of chemical concentrations. At heart it is a very simple study, effectively building on reviews written 15 years ago, but with new data from just a single geographical region. The paper throws a lot of different methods (principal component analysis, enrichment factors , ternary diagrams) at the data. Despite a comparatively long paper, the findings of the study are really nothing new and rather obvious: sea salt ions are closely related and at higher concentrations near the coast; ions such as Ca are mainly terrestrial, sulfate has a marine biogenic source.

I find myself a little torn as to what to recommend for this paper. The data are clearly rare in the sense that there are few data from this sector and from inland sites in general other than Vostok, Dome C and a couple of other sites. The authors do understand previous work and have presented their data in the light of that work. However the whole study is very hampered by the unfortunate fact that surface samples (3 cm surface skims) are just really unsuitable for understanding the chemical climate, and despite the work on the pit samples, it is the surface samples that are the bulk of the paper (7 of 12 data figures plus both tables). Because surface samples inevitably do not cover a full year, it is hard to know what they reveal: at least in some cases they give a misleading impression. For example all the surface samples are collected in the summer, when generally sea salt is high and sulfate is low. Surface samples collected in winter might give a completely different impression, but the paper doesn't make this clear. Surface samples probably represent only a single snowfall, so comparing them from year to year with any seasonal weather statistic (as in Figure 13) is not appropriate.

In the end, I want to be generous and say that the data deserve to be published but a more modest paper is needed, in which the shortcomings of the sample set are more clearly explained, and the paper is stripped down to a shorter length (with perhaps 6 figures and the rest removed or at least moved into the supplement). I don't think the paper brings much insight but it would be a shame if the data were not made available in the peer-reviewed literature so I will recommend major revision, and highlight below where I think the paper can be shortened.

Comments on the text:

Abstract is largely OK, but will need to be shortened in line with the text. The section about ternary diagrams is not needed in the abstract as it adds little to the rest of the text, and the part about SIOL should be excluded.

Line 33/34 "In the interior areas, the negative $nssSO_4^{2-}$ signal in winter snow resulted from inputs of sea salts being completely swamped by the contribution of marine biogenic emission": this doesn't quite make sense. What you mean is that there are high (positive) nss sulfate in inland snow because of marine biogenic sulfate. You have no data on whether negative values would have been seen, so the current wording serves to confuse the reader.

The introduction is generally quite good and the English (with a couple of exceptions that will be picked up in proofreading) is fine.

Methods: After line 154, it should be mentioned that 3 cm at a density of 0.33 as assumed elsewhere is only 1 cm water equivalent (compare snow accumulation in Fig 2a) and therefore the surface samples represent at best a summer sample and in many cases probably a single snowfall.

Fig 2 and section 3.1 already illustrates the problem with the study: there is huge variation at a single site between years, and yet you have no idea whether this variation reflects changes from year to

year, or from week to week within a year. Given that the samples are inevitably collected over a period of days to weeks within a year, much of the spatial variability can arise from temporal change in practice. This doesn't completely invalidate the work but it should be explained.

Fig 3b. I don't see the value of packaging all the samples into a single wheel like this. Firstly how is the calculation done: do you add the concentrations from each site (thus giving more weight to the samples with high concentrations) or is each site normalised before averaging? But wouldn't it be more interesting to show this wheel separately for groups of samples, eg <200 km from coast, 200-600 km, and >600 km. Then you could discuss in a holistic way how the composition changes as you go inland.

Fig 4. Please state in the caption the year the pits were sampled. I know it's somewhere in the text but it's needed here.

Fig 5 can be removed – it adds nothing, and it is sufficient just to say that the accumulation rate is too low at P3 for seasonal variability to be apparent. If necessary simply point out the relevant section in Fig 4.

Fig 6 is unnecessary. Because it pools data from different sites and years the statistics shown really have no meaning. If you really like it, please put it in the supplement.

Section 4.2: PC1 which is clearly sea salt is fine. However it is then rather obvious that Ca as a terrestrial ion, ammonium and sulfate fall into other PCs, and this can be said much more briefly. Personally I think Table 2 is sufficient and Table 1 adds nothing, but I don't insist on losing it.

Fig 9 and 12 and the associated text seem to add little, and they should go into the supplement of be deleted.

Fig 13 and the associated text are very misleading. The three traverses show different values for different sampling dates, but this could be day to day, week to week, month to month or year to year variability and trying to associate with seasonal indices is not relevant. To carry out any such analysis you'd have to calculate the index for the precise dates of the snowfall the surface samples represent. This section should be removed, as should supplement Fig S3.

The other supplement Figures (S1 and S2) also don't add to the message and should not be included (S1 is essentially identical to Fig 4a with the addition of ammonium, and Fig S2 shows information that is already visible in Fig 4a.

---

## Referee Comment (RC2) · Legrand Michel (Referee) · 7 Nov 2020

Review of the manuscript "Spatial and temporal variations in snow chemistry along a traverse from coastal East Antarctica to the ice sheet summit (Dome A)" by *Shi and co-workers*.

This manuscript reports on analysis of inorganic ions in snow samples collected in the frame of intensive program of snow sampling made along several successive traverses achieved from the coast (Zhongshan Station) to Dome A (East Antarctica). The samplings include 594 surface snow samples (upper 3 cm), and 3 snow-pits (down to 1-2 m depth). It is shown that $Cl^-$, $Na^+$, $K^+$, and $Mg^{2+}$ concentrations are high within the narrow coastal region, dropping further inland, while $NO_3^-$ exhibits an opposite trend. No clear spatial trends were found for $SO_4^{2-}$, $NH_4^+$, and $Ca^{2+}$. Data are discussed with respect to potential origin of ions including for minor ones like calcium, ammonium and potassium.

Overall evaluation: First, the authors have to be congratulated for having successfully conducted such a very large inter-annual snow-sampling program, likely sometimes done under harsh weather conditions.

Whereas these data certainly contain interesting information that would be relevant for the Cryosphere journal, inherent to the poor representativeness of 3 cm snow sampling, an in depth interpretation of data is often very difficult and I find that, in the present version of the manuscript, the authors over-interpret them. The manuscript also reveals several misleading presentations and discussions of data. Finally some key previous works are not adequately referenced in the manuscript.

In conclusion I recommend major revisions and in the following I try to identify what can be removed from the manuscript and reversely what can be developed, in particular (and if possible) taking advantage of more information derived from the snow pit data.

**Overall comments:**

**1.** It would be far more logic to first present and discuss the snow pit data showing how large is the seasonal variation of all species including ammonium, nitrate, magnesium, calcium and potassium and discuss the basic causes of that. In the present version, only the well-known species (sodium and sulfate) data in the snow-pit are shown in the main text, whereas ammonium is only seen in the SI. Reporting first (and for all species including ammonium, potassium, calcium, nitrate) on the snow-pit data would permit the readers to appreciate the seasonal variability related to surface snow variability.

**2.** Instead of using enrichment factors (EFs), more illustrative is the calculation of concentration in excess with respect to the seawater composition done with calculation of error propagation (specially for minor species like potassium, magnesium, and calcium). These data would certainly be more useful than EF values to discuss snow pit data (levels, seasonality, etc).

**3.** Section 4.4: The value of this discussion is very weak since based on samples of poor seasonal representativeness. Please remove it (see also my comment below for section 4.4).

**Comments:**

**Title:** please specify "surface snow" chemistry

**Abstract:** Please remove the last sentence since, in no way, your data can demonstrate something (statistically thought) on this topic: « The interannual variations in ion concentrations in surface snow on the traverse are likely linked to the changes in the Southern Indian Ocean low (SIOL) from year to year, and the deepening of the SIOL in summer tends to promote the transport of marine aerosols to Princess Elizabeth Land. »

Also your statement « Snow $NH_4^+$ is mainly associated with marine biological activities » is not really demonstrated (at least as the manuscript stands, see more comments below).

Finally please reword the confusing sentence: «the negative $nssSO_4^{2-}$ signal in winter snow resulted from inputs of sea salts being completely **swamped** by the contribution of marine biogenic emissions. I don't understand what you mean here: do you mean that you have never pure winter snow because of wind mixing after deposition ???

**Introduction:** Several times the choices of your references are strange:

Line 61-65: You miss to cite here Legrand and Delmas (1985) here for a traverse in Adelie Land. This reference is particularly important since it is one the unique traverse for which acidity had been measured (not calculated), see my comment below.

Line 78: I don't think that Saltzman (1995) for the statement "that sulfate in the snow is mainly from marine biogenic sulphur species » is the adequate reference. Please here cite the review from Legrand (1995) or Legrand (1997).

Line 79-80: The two cited references are fine but there are numerous previous works done on that and I would suggest mentioning the article in Nature 1987 (Legrand and Delmas, 1987) for instance.

Line 82-85 : I don't think there is something on the Keene paper on calcium in snow. I suggest citing the study of the Vostok ice core in which the origin (and calculation) of excess potassium, magnesium and calcium were discussed (Legrand et al., 1988).

Line 92: Please cite the first study of ammonium and discussion of its marine origin in Antarctica by Legrand et al. (2000).

Lines 100-105: Please cite Weller et al. (2011) (see my comment on section 4.4).

Legrand M., Sulphur-derived species in polar ice: A review, In *NATO ASI Ser. "Ice cores studies of Global biogeochemical cycles"*, R. Delmas ed., 91-119, 1995.

Legrand, M., Ice-core records of atmospheric sulphur, *Phil. Trans. R. Soc. Lond. B*, 352, 241-250, 1997.

Legrand M., and R.J. Delmas, A 220-year continuous record of volcanic $H_2SO_4$ in the Antarctic ice sheet, *Nature*, 327, 671-676, 1987.

Legrand M., C. Lorius, N.I. Barkov, and V.N. Petrov, Vostok (Antarctica) ice core: Atmospheric chemistry changes over the last climatic cycle (160,000 years), *Atmos. Environ.*, 22, 317-331, 1988.

Legrand M., and C. Saigne, Formate, acetate and methanesulfonate measurements in antarctic ice: Some geochemical implications, *Atmos. Environ.*, 22, 1011-17, 1988.

Legrand, M., and P. Mayewski, Glaciochemistry of polar ice cores: A review, *Reviews of Geophysics*, 35, 219-243, 1997.

Legrand, M., E. Wolff, and D. Wagenbach, Antarctic aerosol and snowfall chemistry: Implications for deep Antarctic ice core chemistry, *Ann. Glaciol.*, 29, 66-72, 2000.

**Section 2.2:**

Line 197-189: This statement "In Antarctic snow, concentrations of $H^+$ are usually not measured directly, but deduced from the ion-balance disequilibrium in the snow » is wrong and very misleading. In fact more than 1000 Antarctic snow and ice samples covering various time periods (present-climate, last glacial age) and collected at various places were measured for $H^+$ (Legrand, 1987; Legrand and Delmas 1984) including along a traverse in Adelie Land (Legrand and Delmas, 1985). From that is was shown that the measurement of chloride, nitrate, sulfate, proton, sodium, ammonium, potassium, magnesium and calcium permit to verify the good balance between measured anions and measured cations. And from these studies that it was

postulated that if not available the H+ concentration can be derived from the equation $H^+] = [SO_4^{2-}] + [NO_3^-] + [Cl^-] - [Na^+] - [NH_4^+] - [K^+] - [Mg^{2+}] - [Ca^{2+}]$ .
So please modify this section accordingly.

Legrand M., and R. Delmas, The ionic balance of antarctic snow: A 10-year detailed record, *Atmos. Environ.*, 18, 1867-1874, 1984.
Legrand M., and R. Delmas, Spatial and temporal variations of snow chemistry in Terre Adélie (East Antarctica), *Ann. Glaciol.*, 7, 20-25, 1985.
Legrand M., Chemistry of Antarctic snow and ice, *J. de Phys.*, 48, 77-86, 1987.

**Section 2.3:**
Why do you play with EF instead of the amount of species present in excess with respect to the seawater composition. The calculations of excess are far more useful to discuss data (see further comments). In any case, calculations of error propagation are clearly needed here, especially for potassium, calcium, and magnesium.

**Sections 2.4:**
In your case (and it is often the case) the PCA approach does not give more information than those that can be simply derived by checking your plots. Checking your Figure 2, it immediately appears that (as expected) you have more sea-salt at the coast than inland (leading to your PC1). Outside of that, other information derived from the PCA analysis are not very powerful (see my further comments).

**Section 3.1:**
Line 234-245: It is for a very short time that melted snow is under saturated with respect to $CO_2$. In fact after 10 min or so the equilibrium is reached but don't forget that another important factor is the temperature (colder is water more $CO_2$ is dissolved). Another source of uncertainty here is the $PCO_2$ in the lab of analysis (related for instance to the number of people). Please report the temperature at which your pH measurements were done.

Would be good here to compare your calculated $H^+$ not only with previous similar estimates but also with previous actual measurements (Legrand and Delmas, 1985, for instance).

**Section 3.2:**
Line 274: I feel that after having calculated $nssSO_4$ (excess sulfate) you will identify an increasing trend of excess-sulfate from the coast to inland due to dry deposition (the sulfate one being obscured by the large amount of sea salt at the coast).

**Section 3.3:**
This section needs to be significantly developed. First please show all species, second calculate excesses and corresponding error bars.

**Section 4.1:**
Please present excess here.
Line 371-373: I disagree with that since only a very small amount of potassium from dust is leachable (and measured with your IC) but it is not at all true for calcium (see Legrand et al., 1988, for instance). Therefore you cannot compare your snow data with the mean crust composition from Bowen that refers to total potassium (insoluble and soluble). Please check in Legrand et al. (1988) or Legrand (1987) information on excess potassium versus excess calcium.

I find also that your calcium data (Fig 2) are often above 4 ppb (that is also higher than seen in numerous ice cores under present-day climate. Please comment.

Line 445-454: I disagree with this discussion. Whereas I agree nitrate is clearly related to atmospheric oxidant, the link to sulfate (its similar presence in PC3) is due to the fact that you concentrate $nssSO_4$ due to dry deposition at sites with low accumulation rate while nitrate is enhanced for a totally different reason (photochemistry).

**Section 4.2:**

Line 430: Your argument of an absence of correlation between ammonium and organic tracers (Shi et al., 2019)  is not correct since the authors invoked a decrease of levoglucosan from the coast to inland due to its photochemical degradation.

Line 433: Checking your Fig S1 in fact you have an outstanding value (17 ppb). Discarding this value I have difficulty to identify a seasonal cycle. Also your mean value (removing the outlier) is close to 2 ppb (it is slightly but significantly higher than what was seen in previous ice core studies). Please comment.

**Section 4.4:** Given the poor representativeness of snow samples, it is clear that examination with respect to SOI is very difficult. For your information, based on a continuous record of 25 year of aerosol, Weller et al. (2011) examine the inter-annual variability with respect to climate-related indices (SAM, SOI, SIE) and nothing very significance had appear.

Weller, R., D. Wagenbach, M. Legrand, C. Elsässer, X. Tian-Kunze, and G. König-Langlo, Continuous 25-years aerosol records at coastal Antarctica: I. Inter-annual variability of ionic compounds and links to climate indices, *Tellus*, 63B, 901-919, DOI:10.1111/j.1600-0889.2011.00542.x, 2011.

End of the review.

---

## Referee Comment (RC3) · Anonymous Referee #1 · 9 Nov 2020

Please note that when I wrote "For example all the surface samples are collected in the summer, when generally sea salt is high and sulfate is low.", I meant the opposite "For example all the surface samples are collected in the summer, when generally sea salt is low and sulfate is high."

---

## Author Comment (AC1) · 7 Dec 2020

**Referee #1**

We thank the reviewer very much for the thoughtful and thorough review of our manuscript. The very helpful comments and suggestions have greatly improved the quality of this paper. Below, we give a point-by-point response to the comments and suggestions of the reviewer, in the order of (1) comments from Referees, (2) author's response, and (3) author's changes in manuscript (referee comments in black; author's response and changes in manuscript in blue).

**(1) comments from Referees**

This paper uses surface samples from a series of traverses between the coast and the inland plateau Dome A station, along with 3 pit sequences, to study the geographic pattern of chemical concentrations. At heart it is a very simple study, effectively building on reviews written 15 years ago, but with new data from just a single geographical region. The paper throws a lot of different methods (principal component analysis, enrichment factors, ternary diagrams) at the data. Despite a comparatively long paper, the findings of the study are really nothing new and rather obvious: sea salt ions are closely related and at higher concentrations near the coast; ions such as Ca are mainly terrestrial, sulfate has a marine biogenic source.

I find myself a little torn as to what to recommend for this paper. The data are clearly rare in the sense that there are few data from this sector and from inland sites in general other than Vostok, Dome C and a couple of other sites. The authors do understand previous work and have presented their data in the light of that work. However the whole study is very hampered by the unfortunate fact that surface samples (3 cm surface skims) are just really unsuitable for understanding the chemical climate, and despite the work on the pit samples, it is the surface samples that are the bulk of the paper (7 of 12 data figures plus both tables). Because surface samples inevitably do not cover a full year, it is hard to know what they reveal: at least in some cases they give a misleading impression. For example all the surface samples are collected in the summer, when generally sea salt is low and sulfate is high. Surface samples collected in winter might give a completely different impression, but the paper doesn't make this clear. Surface samples probably represent only a single snowfall, so comparing them from year to year with any seasonal weather statistic (as in Figure 13) is not appropriate.

In the end, I want to be generous and say that the data deserve to be published but a more modest paper is needed, in which the shortcomings of the sample set are more clearly explained, and the paper is stripped down to a shorter length (with perhaps 6 figures and the rest removed or at least moved into the supplement). I don't think the paper brings much insight but it would be a shame if the data were not made available in the peer-reviewed literature so I will recommend major revision, and highlight below where I think the paper can be shortened.

**(1) author's response**

We greatly appreciate the reviewer for the thoughtful comments of our work.

**(1) author's changes in manuscript**

Following the reviewer's comments, we substantially revised the manuscript. Please see the revised manuscript.

**(2) comments from Referees**

Comments on the text:
Abstract is largely OK, but will need to be shortened in line with the text. The section about ternary diagrams is not needed in the abstract as it adds little to the rest of the text, and the part about SIOL should be excluded.

**(2) author's response**

We agree that the section about the ternary diagrams adds little to the rest of the text, e.g., the section about the excess $Cl^-$ ($nssCl^-$) with respect to the seawater composition. In addition, we agree with both reviewer#1 and reviewer#2 that the section about the factors controlling the interannual variations in chemical ions in surface snow is not robust, considering that the sampled surface ~3cm along the traverse at different locations could represent different periods of snow accumulation, possibly weeks to months or even only a single snowfall event. Thus, the section about the interannual variation and its potential relation to the SIOL was removed in the revised manuscript.

**(2) author's changes in manuscript**

In the abstract section, the text about ternary diagrams and interannual variations of ions in surface snow (the Southern Indian Ocean low (SIOL)) were excluded. And the abstract was also shortened. Please see the revised manuscript. Abstract now reads,
"There is a large variability in environmental conditions across the Antarctic ice sheet, and it is of significance to investigate the snow chemistry at as many locations as possible and over time, given that the ice sheet itself, and precipitation and deposition patterns and trends are changing. The China inland Antarctic traverse from coastal Zhongshan Station to the ice sheet summit (Dome A) covers a variety of environments, allowing for a vast collection of snow chemistry conditions across East Antarctica. Surface snow (the upper ~3 cm, mainly representing the summertime snow) and snow pit samples were collected on this traverse during five campaigns, to comprehensively investigate the spatial and temporal variations in chemical ions ($Cl^-$, $NO_3^-$, $SO_4^{2-}$, $Na^+$, $NH_4^+$, $K^+$, $Mg^{2+}$, and $Ca^{2+}$) and the related controlling factors. Results show that spatial patterns of ions in surface snow are consistent among the five campaigns, with $Cl^-$, $Na^+$, $K^+$, and $Mg^{2+}$ decreasing rapidly with distance from the coast and $NO_3^-$ showing an opposite pattern. No clear spatial trends in $SO_4^{2-}$, $NH_4^+$, and $Ca^{2+}$ were found. In the interior areas, an enrichment of $Cl^-$ versus $Na^+$ with respect to seawater composition is ubiquitous as a result of the deposition of HCl, and $nssCl^-$ (nss, non-sea-salt fraction) can account for up to ~40 % of the total $Cl^-$ budget, while $nssK^+$ and $nssMg^{2+}$ are mainly associated with terrestrial particle mass. On average, $nssCa^{2+}$ and $nssSO_4^{2-}$ in surface snow account for ~77 and 95 % of total $Ca^{2+}$ and total

$SO_4^{2-}$, respectively. The high proportions of the non-sea-salt fractions of $Ca^{2+}$ and $SO_4^{2-}$ are mainly related to terrestrial dust inputs and marine biogenic emissions, respectively. Snow $NH_4^+$ is mainly associated with marine biological activities, with slightly higher concentrations in summer than in winter. On the coast, parts of the winter snow are characterized with negative $nssSO_4^{2-}$ values, and a significant negative correlation between $nssSO_4^{2-}$ and $Na^+$ in wintertime snow was found, suggesting that sea salts originated from the sea ice. In the interior areas, marine biogenic $SO_4^{2-}$ still dominated snow $SO_4^{2-}$ in winter, leading to significant positive $nssSO_4^{2-}$ values. Ion flux assessment suggests an efficient transport of $nssSO_4^{2-}$ to at least as far inland as the ~2800 m contour line."

**(3) comments from Referees**

Line 33/34 "In the interior areas, the negative nssSO42- signal in winter snow resulted from inputs of sea salts being completely swamped by the contribution of marine biogenic emission": this doesn't quite make sense. What you mean is that there are high (positive) nss sulfate in inland snow because of marine biogenic sulfate. You have no data on whether negative values would have been seen, so the current wording serves to confuse the reader.

**(3) author's response**

   Thanks for this comment. The observation suggests that there are high $nssSO_4^{2-}$ values in inland snow during the wintertime, and indeed we have not observed the negative values of $nssSO_4^{2-}$ in interior areas. Following the reviewer's comment, we have reworded this sentence.

**(3) author's changes in manuscript**

This sentence was reworded, and it now reads,
"In the interior areas, marine biogenic $SO_4^{2-}$ still dominated snow $SO_4^{2-}$ in winter, leading to significant positive $nssSO_4^{2-}$ values. "
   For the changes, please see the revision-tracked version of manuscript, section **abstract.**

**(4) comments from Referees**

The introduction is generally quite good and the English (with a couple of exceptions that will be picked up in proofreading) is fine.

**(4) author's response**

   Thanks for the positive comment.

**(4) author's changes in manuscript**

 Several minor changes were made, including adding some previous important references. Please see

the revised version.

**(5) comments from Referees**

Methods: After line 154, it should be mentioned that 3 cm at a density of 0.33 as assumed elsewhere is only 1 cm water equivalent (compare snow accumulation in Fig 2a) and therefore the surface samples represent at best a summer sample and in many cases probably a single snowfall.

**(5) author's response**

We agree with the reviewer and thanks for the constructive comment. Indeed, since the snow accumulation rate varies, the same 3-cm interval, corresponding to about 1 cm water equivalent assuming snow density of 0.33 g cm$^{-3}$, would represent different lengths of time at different locations. At locations with high snow accumulation rates (e.g. the coastal region), the upper 3 cm of snow mainly represents deposition from a short period of the summer season (e.g., on the time scale of weeks). In the most inland Dome A region which has the lowest accumulation rate (6-7 cm snow per year), the upper 3 cm of snow may represent deposition over several months. Still, the information contained in the snow roughly indicates summertime conditions, considering the sampling date of late January to February. In this case, it is reasonable to use the sampled surface snow (~3cm) to investigate the spatial patterns and main origins of chemical ions under summertime conditions. While we accept that the sampling protocol could lead to an imperfect interpretation of that data, as both reviewers raised, collection of the samples covering the same time intervals, in practice, is rather challenging considering the significant variability in surface snow accumulation rate, strong snow drift, etc. Accordingly, we use the surface 3 cm of snow to investigate the overall summertime atmospheric conditions.

**(5) author's changes in manuscript**

In the revised manuscript, we added a paragraph to clarify the representativeness of the surface samples, and it reads,

"It is noted the surface snow represents different lengths of time at different locations, considering the wide range of snow accumulation rates on the traverse (Fig. 2(a)). At locations with high snow accumulation rate on the coast, the upper 3 cm of snow may represent deposition from a few weeks, while the surface 3 cm of snow could represent deposition over a few months on Dome A plateau. Also, it is possible that the upper 3 cm of snow can be representative of a single snowfall. Still, the information contained in the surface snow generally indicates summertime conditions, as the sampling took place during late January and February in each season. This allows for an investigation of summer snow chemistry patterns on the traverse."

**(6) comments from Referees**

Fig 2 and section 3.1 already illustrates the problem with the study: there is huge variation at a single

site between years, and yet you have no idea whether this variation reflects changes from year to year, or from week to week within a year. Given that the samples are inevitably collected over a period of days to weeks within a year, much of the spatial variability can arise from temporal change in practice. This doesn't completely invalidate the work but it should be explained.

**(6) author's response**

We agree with the reviewer that the significant variations in ions in surface snow could arise from the temporal changes in chemical ions. Indeed, there is great variation of ions between years at a single site (Figure 3 in the revised manuscript), and the ion concentrations can vary among samples in the same season (shaded areas in Figures 3 (a) and (b) in the revised manuscript).

Together with the comments from Reviewer#2, the Results section was re-organized and substantially revised. At first, concentrations of all species in snow pits (including $Cl^-$, $NO_3^-$, $SO_4^{2-}$($nssSO_4^{2-}$), $Na^+$, $NH_4^+$, $K^+$, $Mg^{2+}$, and $Ca^{2+}$) were presented in section 3.1. The snow pit data show that there are significant variations in ion concentrations, and even in the same season, ion concentrations can also vary among snowfall events at a specific site. In sections 3.2 and 3.3, ion concentrations in surface snow were presented, and the spatial variability of ions was also included. In addition, in the revised version, it was clarified that surface snow mainly represents summertime deposition, and accordingly the spatial patterns of ions on the traverse can generally represent the summertime conditions.

**(6) author's changes in manuscript**

Following the both reviewers' suggestion, the Results section was re-written, please see the revised manuscript.

**(7) comments from Referees**

Fig 3b. I don't see the value of packaging all the samples into a single wheel like this. Firstly how is the calculation done: do you add the concentrations from each site (thus giving more weight to the samples with high concentrations) or is each site normalised before averaging? But wouldn't it be more interesting to show this wheel separately for groups of samples, eg <200 km from coast, 200-600 km, and >600 km. Then you could discuss in a holistic way how the composition changes as you go inland.

**(7) author's response**

Thanks for the suggestion. In the previous version, the contribution percentages of each ion to the total were calculated from the average values (i.e., the percentages) of all sites on the traverse. Following the reviewer's suggestion, the contribution percentages of each ion to the total ion budget were calculated in different regions. In combination of with the spatial patterns of ion concentrations and previous study (Shi et al., 2018), the traverse was divided into three regions, i.e., the coastal 200km, 200-800km, and the Dome A plateau (800km-Dome A), and then the contribution percentages were

calculated in individual regions. Indeed, there are significant variations in the contribution percentages, with increasing values of $H^+$ and $NO_3^-$ towards inland and an opposite pattern for $Cl^-$, $Na^+$, $Mg^{2+}$, and $NH_4^+$. On the whole traverse, high contribution percentage of $H^+$ is observed, agreeing with previous investigations (Udisti et al., 2004; Traversi et al., 2009; Pasteris et al., 2014), and suggesting acidic characteristics of surface snow.

**(7) author's changes in manuscript**

Following the reviewer's suggestion, this section was re-written and Figure 4 was re-drawn. It now reads,

"The percentages of each constituent to the total ions in surface snow on the traverse are shown in Figs. 4(b)-(d). The most abundant species is $H^+$, accounting for about 30-40 % of the total ions, followed by $NO_3^-$, $SO_4^{2-}$, and $Cl^-$. In general, ions $NH_4^+$, $K^+$, $Mg^{2+}$, and $Ca^{2+}$ are the smallest component of the ionic composition, with the four cation summing to ~5 % of the total. Spatially, the contribution percentages of $H^+$ and $NO_3^-$ increase with increasing distance from the coast, with the highest values on Dome plateau (42.3 and 34.5 %, respectively), while $Cl^-$, $Na^+$, $Mg^{2+}$, and $NH_4^+$ show an opposite pattern and no clear trend was observed for $SO_4^{2-}$. The high contribution percentage of $H^+$ is consistent with previous investigations (Udisti et al., 2004; Traversi et al., 2009; Pasteris et al., 2014), and suggests acidic characteristics of summertime surface snow."

[Figure]

**Figure 4.** Major ions in surface snow on the Chinese inland Antarctic traverse. Concentrations of $H^+$ derived from pH versus those from the ion balance method are shown in panel (a), and contribution percentages of each ion to the total in different regions on the traverse are shown in panels (b)-(d), in

μeq L$^{-1}$. The percentages of each ion in individual regions were calculated from the averages of all sites within the region.

**(8) comments from Referees**

Fig 4. Please state in the caption the year the pits were sampled. I know it's somewhere in the text but it's needed here.

**(8) author's response**

   Thanks for the helpful comment, and the sampling year of each snow pit was included in the caption of Figure 3 (the revised version).

**(8) author's changes in manuscript**

The caption of Figure 3 now reads,
"**Figure 3.** Profiles of chemical ions in snow pits P1 (a), P2 (b), and P3 (c). Snow pits P1 and P2 were sampled in the summer season in 2015-2016, and P3 was sampled in January 2010. The ratios of $SO_4^{2-}/Na^+$ in snow samples were also present. Red arrows in panels (a) and (b) represent the middle of the identified summer, and shaded areas denote summer seasons (see text). The red dashed line in panel (a) represents the ratio of $SO_4^{2-}/Na^+$ in bulk seawater, while the red dashed line in panel (c) signifies the first snow sample significantly influenced by the Pinatubo eruption. One seasonal cycle generally represents local $Na^+$ minima and $nssSO_4^{2-}$ and $SO_4^{2-}/Na^+$ maxima."

**(9) comments from Referees**

Fig 5 can be removed – it adds nothing, and it is sufficient just to say that the accumulation rate is too low at P3 for seasonal variability to be apparent. If necessary simply point out the relevant section in Fig 4.

**(9) author's response**

   We agree with the reviewer, and Figure 5 (in original version) was removed. In the revised manuscript, the figure was included in the supplementary materials (Figure S1), to show the variations of ions in the years 2008 and 2009.

**(9) author's changes in manuscript**

   Figure 5 was removed, and the section was revised accordingly. It now reads,
   "At P3, $Na^+$, $nssSO_4^{2-}$ and the ratios of $SO_4^{2-}/Na^+$ fluctuate significantly (Fig. 3(c)), and these contrasts are unlikely indicative of the seasonal cycles as that for P1 and P2. In a full year of snow accumulation at P3, on average, about 7-8 samples were collected, allowing for examining the seasonal

variability of ions. Following the field measurements of snow accumulation rate at Dome A during 2008-2011 (~20 kg m$^{-2}$ a$^{-1}$; Ding et al., 2015), the snow samples covering the years 2008 and 2009 can be roughly identified, assuming an even distribution of snow accumulation throughout the year. In total, there are 7 and 8 samples identified in the years 2008 and 2009, respectively (Fig. S1), and no seasonal cycles in Na$^+$, nssSO$_4^{2-}$, and SO$_4^{2-}$/Na$^+$ ratio were found due to the low snow accumulation rate at P3. In addition, the post-depositional processes (e.g., migration, diffusion, and ventilation processes) and/or wind scouring can obscure the original signal (Cunningham and Waddington, 1993; Albert and Shultz, 2002; Libois et al., 2014; Caiazzo et al., 2016), resulting in the absence of seasonal cycles of ions at P3."

**(10) comments from Referees**

Fig 6 is unnecessary. Because it pools data from different sites and years the statistics shown really have no meaning. If you really like it, please put it in the supplement.

**(10) author's response**

Agree with the reviewer. Together with the comments of Reviewer#2, the calculation of Enrichment Factors (EFs) was excluded in the revised manuscript. Instead, the calculation of concentration in excess with respect to the seawater composition was included (i.e., the non-sea-salt fractions of ions). Accordingly, in the discussion section, the EFs of chemical ions were replaced with the excess concentrations.

**(10) author's changes in manuscript**

Figure 6 in the original version was removed. Please see the revised manuscript.

**(11) comments from Referees**

Section 4.2: PC1 which is clearly sea salt is fine. However it is then rather obvious that Ca as a terrestrial ion, ammonium and sulfate fall into other PCs, and this can be said much more briefly. Personally I think Table 2 is sufficient and Table 1 adds nothing, but I don't insist on losing it.

**(11) author's response**

We agree with the reviewer that PC1 is clearly indicative of the sea salt inputs, while SO$_4^{2-}$, NH$_4^+$, and Ca$^{2+}$ fall into other components suggest different main sources. Following the reviewer's suggestion, this section was shortened and the main sources of ions in surface snow were discussed briefly. Table 1 shows the main outcomes of PCA, especially the loadings of individual ions in each PC, and thus this table was still included in the updated manuscript. Then, one can easily read the loadings of individual ions and the contribution percentages of each component to the total variance. Table 2 shows the correlation coefficients of chemical ions in surface snow, together with the PCA results

(Table 1), the main sources of ions can be clearly distinguished.

**(11) author's changes in manuscript**

In Section 4.2, the different groups of ions and the possible sources were discussed briefly, and the section was shortened. In the revised manuscript, both tables 1 and 2 were included, in order to show PCA and correlation analysis results clearly. The revised text now reads,

"PC1 accounts for 46 % of the variance and is highly loaded by $Cl^-$, $Na^+$, $K^+$, and $Mg^{2+}$, with the factor loadings higher than 0.7. In addition, the four species are correlated well with each other (Table 2), suggesting the variation of the four species is dominated by sea salt aerosols. PC2 accounts for 17 % of the total variance, and the loading values of $NH_4^+$ and $Ca^{2+}$ in PC2 are high, ~0.8. $Ca^{2+}$ is mainly from terrestrial particle mass, while $NH_4^+$ is thought to be mainly associated with biological decomposition of organic matter in the Southern Ocean (Johnson et al., 2007; Kaufmann et al., 2010). In addition, biomass burning from mid-latitudes can contribute to snow $NH_4^+$ at some sites (Pasteris et al., 2014), and the penguin colony emissions can be important inputs to $NH_4^+$ in snow several km from the colony (Rankin and Wolff, 2000). On this traverse, no correlation was found between $NH_4^+$ and biomass burning tracers (e.g., black carbon and phenolic compounds) in surface snow (Shi et al., 2019; Ma et al., 2020), suggesting a minor role of biomass burning emissions. Thus, the high $NH_4^+$ concentrations on the coast are likely associated with marine biogenic emissions. In this case, it is possible that a similar transport pathway (i.e., preferentially transported long distance in free transport; Krinner and Genthon, 2003; Kaufmann et al., 2010; Krinner et al., 2010) can explain, at least in part, the positive loadings of both $NH_4^+$ and $Ca^{2+}$ in PC2.

$NO_3^-$ is highly loaded in PC3, which accounts for 13 % of the system variance. On this traverse, $NO_3^-$ in the snow has been extensively investigated, and it is proposed that $NO_3^-$ concentrations were influenced by post-depositional processing which is largely dependent on snow accumulation rate (Shi et al., 2015; Shi et al., 2018a; Shi et al., 2018b). A negative relationship was found between $NO_3^-$ and snow accumulation rate (Fig. 6(b)), suggesting a high degree of $NO_3^-$ cycling driven by photolysis at low snow accumulation sites.

$SO_4^{2-}$ did not show high loadings in any of the three extracted components. Its positive loading in PC1 (0.55) and weak relationships between $SO_4^{2-}$ and sea salts ($Cl^-$ and $Na^+$) likely supports the contribution of sea salt aerosols, although a minor one. A positive loading of $SO_4^{2-}$ is also present in PC3 (0.42), and a weak correlation was found between $SO_4^{2-}$ and $NO_3^-$. Both $SO_4^{2-}$ (or $nssSO_4^{2-}$) and $NO_3^-$ are negatively correlated with snow accumulation rate (Fig. 6), but with distinct mechanisms. $nssSO_4^{2-}$ can be concentrated due to dry deposition at sites with low snow accumulation rate, while elevated $NO_3^-$ concentrations are linked to the photochemical cycling and re-deposition (discussed above). In addition, $nssSO_4^2$ and $NO_3^-$ are mainly associated with the secondary aerosols, and the production of both species in summer is closely related to the oxidants $HO_x$, $RO_x$, etc (Ishino et al., 2017; Shi et al., 2018a), which may also contribute to the correlation between $SO_4^{2-}$ and $NO_3^-$."

**(12) comments from Referees**

Fig 9 and 12 and the associated text seem to add little, and they should go into the supplement of be

deleted.

We agree with the reviewer that the ternary diagram of $Cl^-$, $Na^+$, and $SO_4^{2-}$, and the related text are in part similar with the results of excess concentrations with respect to the seawater composition. The patterns of $Cl^-$, $Na^+$, and $SO_4^{2-}$ can more clearly characterize the modification processes to sea salt aerosols, in comparison with the excess concentrations alone. In this case, we keep Figure 9 (in the original version) in the revised manuscript, to show the spatial variation in the degree of sea salt modification in surface snow. Figure 12 (in the original version) shows the seasonal variations in the degree of sea salt modification. Considering that the seasonal variations in ion concentrations and excess concentrations with respect to the seawater composition were discussed extensively in section 4.3, figure 12 (in the original version) was moved to the supplementary materials following the reviewer's suggestion. Accordingly, the associated text was significantly shortened.

**(12) author's changes in manuscript**

Figure 12 (in the original version) was moved to the supplementary materials, and the associated text was shortened. It now reads,

"The ternary diagram of $Cl^-$, $Na^+$, and $SO_4^{2-}$ can well characterize the modification processes to sea salt aerosols, and the ternary plot of the three ions in surface snow is shown in Fig. 7. The values of the ions were normalized via the following equation,

$X=[X]/([Na^+]+[Cl^-]+[SO_4^{2-}])$ Eq. (3),

where [X] is the concentration of ion X in the snow (in $\mu eq\ L^{-1}$). The dashed line between the seawater reference value and the $SO_4^{2-}$ vertex represents the sea salt aerosol composition with additional $SO_4^{2-}$, i.e., the ratio of $Cl^-/Na^+$ keeps constant (1.17) with additional $SO_4^{2-}$ along the dashed line. The presence of acids ($HNO_3$ and $H_2SO_4$) would result in the liberation of HCl into the atmosphere via reactions R1 and R2, resulting in the changes in Cl/Na ratios, i.e., either $Cl^-$ loss or gain are located right or left of the line, respectively. It is shown that all of the data points are above the seawater plot, suggesting an enrichment of $SO_4^{2-}$ in surface snow. Most of the data points are located left of the line, indicating the general enrichment of $Cl^-$ due to reactions R1 and R2 occurring in the atmosphere and/or in the snowpack. But the coastal data points are generally close to the line, suggesting that the degree of sea salt modification is generally low in the snow.

Similar to the surface snow, the modification processes to sea salt aerosols is negligible in snow pit P1, while the ubiquitous modification process to sea salts throughout the year was found in the interior areas (P2 and P3; Fig. S3). Thus, $Cl^-$ in inland Antarctica, often deviating from the seawater dilution line remarkably in both summer and winter, is not a quantitative indicator of sea salts in snow."

**(13) comments from Referees**

Fig 13 and the associated text are very misleading. The three traverses show different values for different sampling dates, but this could be day to day, week to week, month to month or year to year

variability and trying to associate with seasonal indices is not relevant. To carry out any such analysis you'd have to calculate the index for the precise dates of the snowfall the surface samples represent. This section should be removed, as should supplement Fig S3.

**(13) author's response**

We agree with the reviewer and thanks for the comment. Together with the comments of Reviewer#2, section 4.4 was removed in the revised manuscript. In addition, Figure S3, which shows the climatological mean sea level pressure distribution over the southern ocean in the austral summers in different years, was also removed.

**(13) author's changes in manuscript**

Section 4.4 was removed. Please see the revised manuscript.

**(14) comments from Referees**

The other supplement Figures (S1 and S2) also don't add to the message and should not be included (S1 is essentially identical to Fig 4a with the addition of ammonium, and Fig S2 shows information that is already visible in Fig 4a.

**(14) author's response**

We agree with the reviewer. The concentration profile of $NH_4^+$ in the snow pit is now included in Figure 3a (in the revised version), and thus Figure S1 was removed. The information of Figure S2 is now present in Figure 3a, and Figure S2 was also deleted.

**(14) author's changes in manuscript**

Figure S1 and S2 were removed

**References**

Bertler N., Mayewski P.A., Aristarain A., Barrett P., Becagli S., Bernardo R., Bo S., Xiao C., Curran M. and Qin D. (2005) Snow chemistry across Antarctica. *Ann. Glaciol.* 41, 167-179.

Duderstadt K.A., Dibb J.E., Schwadron N.A., Spence H.E., Solomon S.C., Yudin V.A., Jackman C.H. and Randall C.E. (2016) Nitrate ion spikes in ice cores not suitable as proxies for solar proton events. *J. Geophys. Res.* 121, 2994-3016.

Erbland J., Vicars W., Savarino J., Morin S., Frey M., Frosini D., Vince E. and Martins J. (2013) Air-snow transfer of nitrate on the East Antarctic Plateau - Part 1: Isotopic evidence for a photolytically driven dynamic equilibrium in summer. *Atmos. Chem. Phys.* 13, 6403-6419.

France J., King M., Frey M., Erbland J., Picard G., Preunkert S., MacArthur A. and Savarino J. (2011) Snow optical properties at Dome C (Concordia), Antarctica; implications for snow emissions and snow chemistry of reactive nitrogen. *Atmos. Chem. Phys.* 11, 9787-9801.

Frey M.M., Savarino J., Morin S., Erbland J. and Martins J. (2009) Photolysis imprint in the nitrate stable isotope signal in snow and atmosphere of East Antarctica and implications for reactive nitrogen cycling. *Atmos. Chem. Phys.* 9, 8681-8696.

Holland P.R., Bruneau N., Enright C., Losch M., Kurtz N.T. and Kwok R. (2014) Modeled Trends in Antarctic Sea Ice Thickness. *J. Climate* 27, 3784-3801.

Marion G., Farren R. and Komrowski A. (1999) Alternative pathways for seawater freezing. *Cold Reg. Sci. Technol.* 29, 259-266.

Mulvaney R., Wagenbach D. and Wolff E.W. (1998) Postdepositional change in snowpack nitrate from observation of year-round near-surface snow in coastal Antarctica. *J. Geophys. Res.* 103, 11021-11031.

Pasteris D., McConnell J.R., Edwards R., Isaksson E. and Albert M.R. (2014) Acidity decline in Antarctic ice cores during the Little Ice Age linked to changes in atmospheric nitrate and sea salt concentrations. *J. Geophys. Res.* 119, 5640-5652.

Rankin A.M., Wolff E.W. and Martin S. (2002) Frost flowers: Implications for tropospheric chemistry and ice core interpretation. *J. Geophys. Res.* 107, 4683.

Savarino J., Kaiser J., Morin S., Sigman D.M. and Thiemens M.H. (2007) Nitrogen and oxygen isotopic constraints on the origin of atmospheric nitrate in coastal Antarctica. *Atmos. Chem. Phys.* 7, 1925-1945.

Shi G., Buffen A.M., Hastings M.G., Li C., Ma H., Li Y., Sun B., An C. and Jiang S. (2015) Investigation of post-depositional processing of nitrate in East Antarctic snow: isotopic constraints on photolytic loss, re-oxidation, and source inputs. *Atmos. Chem. Phys.* 15, 9435–9453.

Shi G., Buffen A.M., Ma H., Hu Z., Sun B., Li C., Yu J., Ma T., An C., Jiang S., Li Y. and Hastings M.G. (2018) Distinguishing summertime atmospheric production of nitrate across the East Antarctic Ice Sheet. *Geochim. Cosmochim. Acta* 231, 1-14.

Wagenbach D., Ducroz F., Mulvaney R., Keck L., Minikin A., Legrand M., Hall J.S. and Wolff E.W. (1998a) Sea-salt aerosol in coastal Antarctic regions. *J. Geophys. Res.* 103, 10961-10974.

Wagenbach D., Legrand M., Fischer H., Pichlmayer F. and Wolff E.W. (1998b) Atmospheric near-surface nitrate at coastal Antarctic sites. *J. Geophys. Res.* 103, 11007-11020.

Wolff E.W., Bigler M., Curran M., Dibb J., Frey M., Legrand M. and McConnell J. (2012) The Carrington event not observed in most ice core nitrate records. *Geophys. Res. Lett.* 39, L08503.

Wolff E.W., Bigler M., Curran M.A.J., Dibb J.E., Frey M.M., Legrand M. and Mcconnell J.R. (2016) Comment on "Low time resolution analysis of polar ice cores cannot detect impulsive nitrate events" by D.F. Smart et al. *J. Geophys. Res.* 121, 1920-1924.

Wolff E.W., Jones A.E., Bauguitte S.-B. and Salmon R.A. (2008) The interpretation of spikes and trends in concentration of nitrate in polar ice cores, based on evidence from snow and atmospheric measurements. *Atmos. Chem. Phys.* 8, 5627-5634.

Zatko M., Grenfell T., Alexander B., Doherty S., Thomas J. and Yang X. (2013) The influence of snow grain size and impurities on the vertical profiles of actinic flux and associated $NO_x$ emissions on the Antarctic and Greenland ice sheets. *Atmos. Chem. Phys.* 13, 3547-3567.

**End of responses to Referee #1.**

---

## Author Comment (AC2) · 7 Dec 2020

We appreciate Referee #1 for updating the comments.

---

## Author Comment (AC3) · 7 Dec 2020

**Reviewer #2 (Prof. Michel Legrand)**

**We are very grateful to Prof. Michel Legrand for his detailed comments and very useful and constructive suggestions. The manuscript has been substantially modified and reformatted based on these comments/suggestions. Below, we give a point-by-point response to the comments and suggestions of Prof. Michel Legrand, in the order of (1) comments from Referees, (2) author's response, and (3) author's changes in manuscript. Prof. Michel Legrand's comments are in black, and the responses are in blue.**

**(1) comments from Referees**

This manuscript reports on analysis of inorganic ions in snow samples collected in the frame of intensive program of snow sampling made along several successive traverses achieved from the coast (Zhongshan Station) to Dome A (East Antarctica). The samplings include 594 surface snow samples (upper 3 cm), and 3 snow-pits (down to 1-2 m depth). It is shown that $Cl^-$, $Na^+$, $K^+$, and $Mg^{2+}$ concentrations are high within the narrow coastal region, dropping further inland, while $NO_3^-$ exhibits an opposite trend. No clear spatial trends were found for $SO_4^{2-}$, $NH_4^+$, and $Ca^{2+}$. Data are discussed with respect to potential origin of ions including for minor ones like calcium, ammonium and potassium.

Overall evaluation: First, the authors have to be congratulated for having successfully conducted such a very large inter-annual snow-sampling program, likely sometimes done under harsh weather conditions.

Whereas these data certainly contain interesting information that would be relevant for the Cryosphere journal, inherent to the poor representativeness of 3 cm snow sampling, an in depth interpretation of data is often very difficult and I find that, in the present version of the manuscript, the authors over-interpret them. The manuscript also reveals several misleading presentations and discussions of data. Finally some key previous works are not adequately referenced in the manuscript.

In conclusion I recommend major revisions and in the following I try to identify what can be removed from the manuscript and reversely what can be developed, in particular (and if possible) taking advantage of more information derived from the snow pit data.

**(1) author's response**

We appreciate Prof. Michel Legrand taking the time to complete the review and welcome the very helpful and constructive comments.

**(1) author's changes in manuscript**

We revised the manuscript substantially following the comments and suggestions, see responses below.

**(2) comments from Referees**

Overall comments:

1. It would be far more logic to first present and discuss the snow pit data showing how large is the seasonal variation of all species including ammonium, nitrate, magnesium, calcium and potassium and discuss the basic causes of that. In the present version, only the well-known species (sodium and sulfate) data in the snow-pit are shown in the main text, whereas ammonium is only seen in the SI. Reporting first (and for all species including ammonium, potassium, calcium, nitrate) on the snow-pit data would permit the readers to appreciate the seasonal variability related to surface snow variability.

**(2) author's response**

We agree with Prof. Michel Legrand and thanks for the helpful suggestion. Indeed, there is significant seasonal variation of ions at P1 and P2 (Figure 3 in the revised manuscript). The Results section 3 was re-organized and substantially revised as suggested by the reviewer. Section 3.1 now discussed the concentrations of all species in snow pits (including $Cl^-$, $NO_3^-$, $SO_4^{2-}$($nssSO_4^{2-}$), $Na^+$, $NH_4^+$, $K^+$, $Mg^{2+}$ and $Ca^{2+}$). The snow pit data show that there are significant variations in ion concentrations, and even in the same summer or winter season, ion concentrations can also vary among snowfall events at a specific site. In sections 3.2 and 3.3, ion concentrations in surface snow are presented, and the spatial variability of ions is included. In addition, in the revised version, it was clarified that surface snow mainly represents summertime deposition, and accordingly the spatial patterns of ions on the traverse can roughly represent summertime conditions.

**(2) author's changes in manuscript**

Section 3 Results was re-organized as suggested by the reviewer, and Figure 3 was reproduced. Please see the revised manuscript.

**(3) comments from Referees**

2. Instead of using enrichment factors (EFs), more illustrative is the calculation of concentration in excess with respect to the seawater composition done with calculation of error propagation (specially for minor species like potassium, magnesium, and calcium). These data would certainly be more useful than EF values to discuss snow pit data (levels, seasonality,etc).

**(3) author's response**

We agree with Prof. Michel Legrand and thanks for the comment. The section of EFs calculation in the original version was removed. In the Result and Discussion sections, the ion concentrations in excess with respect to seawater composition (i.e., the non-sea-salt fractions of ions, nssX, with X of an ion) were used to discuss the levels, seasonality, main sources, etc. Accordingly, all the text associated with the EFs was replaced with the excess concentrations (nssX). Also, the error for the calculation of the non-sea-salt fractions were included, also see the response #(18) below.

**(3) author's changes in manuscript**

For the revisions, please see the revised manuscript sections 2 Methods, 3 Results, and 4 Discussions. Accordingly, the related text in Abstract and Conclusions sections were also revised.

**(4) comments from Referees**

3. Section 4.4: The value of this discussion is very weak since based on samples of poor seasonal representativeness. Please remove it (see also my comment below for section 4.4).

**(4) author's response**

We agree with Prof. Michel Legrand. Indeed, the surface samples, which represent different lengths of time at varied locations, are unlikely representative of seasonal (or interannual) variations. Together with the comments of Reviewer#1, this section was removed.

**(4) author's changes in manuscript**

Section 4.4 (in the original version) was removed. Please see the revised version of the manuscript.

**(5) comments from Referees**

Comments:
Title: please specify "surface snow" chemistry

**(5) author's response**

The "surface snow" was specified in the title.

**(5) author's changes in manuscript**

Now the title reads,
"Spatial and temporal variations in surface snow chemistry along a traverse from coastal East Antarctica to the ice sheet summit (Dome A)"

**(6) comments from Referees**

Abstract: Please remove the last sentence since, in no way, your data can demonstrate something (statistically thought) on this topic: «The interannual variations in ion concentrations in surface snow on the traverse are likely linked to the changes in the Southern Indian Ocean low (SIOL) from year to year, and the deepening of the SIOL in summer tends to promote the transport of marine aerosols to Princess Elizabeth Land. »
Also your statement «Snow $NH_4^+$ is mainly associated with marine biological activities »

is not really demonstrated (at least as the manuscript stands, see more comments below). Finally please reword the confusing sentence: «the negative nssSO42- signal in winter snow resulted from inputs of sea salts being completely swamped by the contribution of marine biogenic emissions. I don't understand what you mean here: do you mean that you have never pure winter snow because of wind mixing after deposition ???

**(6) author's response**

Agreed. Section 4.4 was removed and the abstract as well as conclusions were revised accordingly.

As for the possible sources of $NH_4^+$ in surface snow, several sentences were included in the manuscript (see response below).

The sentence "the negative $nssSO_4^{2-}$ signal in winter snow resulted from inputs of sea salts being completely swamped by the contribution of marine biogenic emissions" was rephrased.

**(6) author's changes in manuscript**

The abstract was revised accordingly, and it now reads,

"There is a large variability in environmental conditions across the Antarctic ice sheet, and it is of significance to investigate the snow chemistry at as many locations as possible and over time, given that the ice sheet itself, and precipitation and deposition patterns and trends are changing. The China inland Antarctic traverse from coastal Zhongshan Station to the ice sheet summit (Dome A) covers a variety of environments, allowing for a vast collection of snow chemistry conditions across East Antarctica. Surface snow (the upper ~3 cm, mainly representing the summertime snow) and snow pit samples were collected on this traverse during five campaigns, to comprehensively investigate the spatial and temporal variations in chemical ions ($Cl^-$, $NO_3^-$, $SO_4^{2-}$, $Na^+$, $NH_4^+$, $K^+$, $Mg^{2+}$, and $Ca^{2+}$) and the related controlling factors. Results show that spatial patterns of ions in surface snow are consistent among the five campaigns, with $Cl^-$, $Na^+$, $K^+$, and $Mg^{2+}$ decreasing rapidly with distance from the coast and $NO_3^-$ showing an opposite pattern. No clear spatial trends in $SO_4^{2-}$, $NH_4^+$, and $Ca^{2+}$ were found. In the interior areas, an enrichment of $Cl^-$ versus $Na^+$ with respect to seawater composition is ubiquitous as a result of the deposition of HCl, and $nssCl^-$ (nss, non-sea-salt fraction) can account for up to ~40 % of the total $Cl^-$ budget, while $nssK^+$ and $nssMg^{2+}$ are mainly associated with terrestrial particle mass. On average, $nssCa^{2+}$ and $nssSO_4^{2-}$ in surface snow account for ~77 and 95 % of total $Ca^{2+}$ and total $SO_4^{2-}$, respectively. The high proportions of the non-sea-salt fractions of $Ca^{2+}$ and $SO_4^{2-}$ are mainly related to terrestrial dust inputs and marine biogenic emissions, respectively. Snow $NH_4^+$ is mainly associated with marine biological activities, with slightly higher concentrations in summer than in winter. On the coast, parts of the winter snow are characterized with negative $nssSO_4^{2-}$ values, and a significant negative correlation between $nssSO_4^{2-}$ and $Na^+$ in wintertime snow was found, suggesting that sea salts originated from the sea ice. In the interior areas, marine biogenic $SO_4^{2-}$ still dominated snow $SO_4^{2-}$ in winter, leading to significant positive $nssSO_4^{2-}$ values. Ion flux assessment suggests an efficient transport of $nssSO_4^{2-}$ to at least as far inland as the ~2800 m contour line."

**(7) comments from Referees**

Introduction: Several times the choices of your references are strange:

Line 61-65: You miss to cite here Legrand and Delmas (1985) here for a traverse in

Adelie Land. This reference is particularly important since it is one the unique traverse for

which acidity had been measured (not calculated), see my comment below.

**(7) author's response**

Thanks. We are sorry for missing some previous important works in the Introduction section.

The reference, Legrand and Delmas (1985) was included in the revised manuscript.

Legrand M. and Delmas R.J. (1985) Spatial and temporal variations of snow chemistry in Terre Adélie

(East Antarctica). Ann. Glaciol. 7, 20-25.

**(7) author's changes in manuscript**

The sentence now reads,

"Snow chemistry has been broadly investigated along traverses during the International

Trans-Antarctic Scientific Expedition (ITASE), e.g., DDU to Dome C, coast-interior traverse in Terre

Adelie, Syowa to Dome F, Terra Nova Bay to Dome C, 1990 ITASE, and US ITASE in West Antarctica

(Legrand and Delmas, 1985; Qin et al., 1992; Mulvaney and Wolff, 1994; Proposito et al., 2002;

Suzuki et al., 2002; Dixon et al., 2013),"

**(8) comments from Referees**

Line 78: I don't think that Saltzman (1995) for the statement "that sulfate in the snow is

mainly from marine biogenic sulphur species »is the adequate reference. Please here cite the

review from Legrand (1995) or Legrand (1997).

**(8) author's response**

Thanks for the point. The references Legrand (1995) and Legrand (1997) were cited here.

Legrand M., 1995. Sulphur-derived species in polar ice: a review, in: Delmas, R. (Ed.), Ice core studies

of global biogeochemical cycles. Springer, pp. 91-119.

Legrand M. (1997) Ice–core records of atmospheric sulphur. Phil. Trans. R. Soc. Lond. B 352,

241-250.

**(8) author's changes in manuscript**

The sentence now reads,

"$SO_4^{2-}$ in the snow is mainly from marine biogenic sulfur species, dimethylsulphide (DMS) (Legrand,

1995; 1997),"

**(9) comments from Referees**

Line 79-80: The two cited references are fine but there are numerous previous works done on that and I would suggest mentioning the article in Nature 1987 (Legrand and Delmas, 1987) for instance.

**(9) author's response**

Agree that there are previous works done on that large volcanic eruption emissions can episodically contribute to the very high concentrations of $SO_4^{2-}$ in ice core. The important work Legrand and Delmas (1987) was included here.

Legrand M. and Delmas R. (1987) A 220-year continuous record of volcanic H2SO4 in the Antarctic ice sheet. Nature 327, 671-676.

**(9) author's changes in manuscript**

The reference was included, and the sentence now reads,

"with a small proportion from sea salt aerosols, while large volcanic eruption emissions can episodically contribute to spikes in $SO_4^{2-}$ concentration (Legrand and Delmas, 1987; Jiang et al., 2012; Cole-Dai et al., 2013)."

**(10) comments from Referees**

Line 82-85 : I don't think there is something on the Keene paper on calcium in snow. I suggest citing the study of the Vostok ice core in which the origin (and calculation) of excess potassium, magnesium and calcium were discussed (Legrand et al., 1988).

**(10) author's response**

Thanks for the point. The reference Keene et al. (2007) was replaced with Legrand et al. (1988).

Legrand M., Lorius C., Barkov N. and Petrov V. (1988) Vostok (Antarctica) ice core: Atmospheric chemistry changes over the last climatic cycle (160,000 years). Atmos. Environ. 22, 317-331.

**(10) author's changes in manuscript**

The reference was changed, and it now reads,

"Terrestrial sources can also contribute to potassium ($K^+$) and magnesium ($Mg^{2+}$) in snow, but the contribution proportion varies significantly among sites (Legrand et al., 1988; Khodzher et al., 2014)."

**(11) comments from Referees**

Line 92: Please cite the first study of ammonium and discussion of its marine origin in Antarctica by Legrand et al. (2000).

**(11) author's response**

We agree with that Legrand et al. (1999) (published in 1999 instead of 2000) is the first study of ammonium and discussion of its marine origin in Antarctica. And this work was cited. Since Pasteris et al. (2014) discussed the ammonium source of biomass burning, the reference Pasteris et al. (2014) was kept here.

Legrand M., Wolff E. and Wagenbach D. (1999) Antarctic aerosol and snowfall chemistry: implications for deep Antarctic ice-core chemistry. Ann. Glaciol. 29, 66-72.

**(11) author's changes in manuscript**

The original references were replaced with Legrand et al. (1999), and the sentence now reads,

"and biogenic emissions in the Southern Ocean and/or mid-latitude biomass burning were proposed to be the major sources, depending on the investigation sites (Legrand et al., 1999; Pasteris et al., 2014)"

**(12) comments from Referees**

Lines 100-105: Please cite Weller et al. (2011) (see my comment on section 4.4).

**(12) author's response**

Thanks for the point, and the work Weller et al. (2011) was included in the section.

Weller R., Wagenbach D., Legrand M., Elsässer C., Tian-kunze X., Königlanglo G. and niglanglo G. (2011) Continuous 25-yr aerosol records at coastal Antarctica-: inter-annual variability of ionic compounds and links to climate indices. Tellus B 63, 901-919.

**(12) author's changes in manuscript**

The section now reads,

"On annual to decadal time scales, ion concentrations in snow and ice tend to be associated with changes in transport from year to year (Severi et al., 2009; Weller et al., 2011), and thus large scale atmospheric and oceanic circulation in the Southern Hemisphere, such as the Southern Annular Mode (SAM), Southern Oscillation (SO) and Southern Indian Ocean Dipole (SIOD), could potentially influence variations in ions in ice (Russell and McGregor, 2010; Weller et al., 2011; Mayewski et al., 2017)."

**(13) comments from Referees**

Section 2.2:
Line 197-189: This statement "In Antarctic snow, concentrations of $H^+$ are usually not measured directly, but deduced from the ion-balance disequilibrium in the snow »is wrong and very misleading. In fact more than 1000 Antarctic snow and ice samples covering various time periods (present-climate, last glacial age) and collected at various places were measured for $H^+$ (Legrand, 1987; Legrand and Delmas 1984) including along a traverse in Adelie Land (Legrand

and Delmas, 1985). From that is was shown that the measurement of chloride, nitrate, sulfate, proton, sodium, ammonium, potassium, magnesium and calcium permit to verify the good balance between measured anions and measured cations. And from these studies that it was postulated that if not available the H+ concentration can be derived from the equation $[H+] = [SO_4^{2-}] + [NO_3^-] + [Cl^-] - [Na^+] - [NH_4^+] - [K^+] - [Mg^{2+}] - [Ca^{2+}]$ .

So please modify this section accordingly.

**(13) author's response**

Agreed and thanks for this comment. Indeed, from a number of observations (e.g., Legrand and Delmas, 1984; Legrand and Delmas, 1985; Legrand, 1987) it is deduced that concentrations of $H^+$ can be reasonably deduced from the ion-balance disequilibrium, if the direct measurements of $H^+$ are unavailable. This section was revised following the reviewer's suggestion.

Legrand M. (1987) Chemistry of Antarctic snow and ice. J. de Phys. 48, 77-86.

Legrand M. and Delmas R.J. (1985) Spatial and temporal variations of snow chemistry in Terre Adélie (East Antarctica). Ann. Glaciol. 7, 20-25.

Legrand M.R. and Delmas R.J. (1984) The ionic balance of Antarctic snow: a 10-year detailed record. Atmos. Environ. 18, 1867-1874.

**(13) author's changes in manuscript**

This section was re-written, and it now reads,

"In Antarctic snow, previous observations suggested that concentrations of $H^+$ can be reasonably deduced from the ion-balance disequilibrium, if the direct measurements of $H^+$ are unavailable (Legrand and Delmas, 1984; Legrand and Delmas, 1985; Legrand, 1987). Here, $H^+$ concentration is calculated as follows.

$[H^+] = [SO_4^{2-}] + [NO_3^-] + [Cl^-] - [Na^+] - [NH_4^+] - [K^+] - [Mg^{2+}] - [Ca^{2+}]$ Eq. (1),"

**(14) comments from Referees**

Section 2.3:

Why do you play with EF instead of the amount of species present in excess with respect to the seawater composition. The calculations of excess are far more useful to discuss data (see further comments). In any case, calculations of error propagation are clearly needed here, especially for potassium, calcium, and magnesium.

**(14) author's response**

We agree that the calculation of non-sea-salt fractions of ions (nssX) (i.e., the concentrations in excess with respect to the seawater composition) is a more illustrative way. In the previous version, the enrichment factors (EFs) is a measurement of whether or not an ion is present in a relative abundance similar to that of seawater, which is an alternative way of showing the enrichment of an ion with respect to the bulk seawater composition. In the revised manuscript, we only focused on the calculation

and discussion of the non-sea-salt fractions of ions in snow, following the reviewer's suggestion.

**(14) author's changes in manuscript**

Section 2.3 was removed, and the associated text in the manuscript was revised accordingly. Also, the errors of non-sea-salt fractions of ions are now presented in the revised manuscript (nssX±1σ).
Please see the revised manuscript, sections 2, 3, and 4.

**(15) comments from Referees**

Sections 2.4:
In your case (and it is often the case) the PCA approach does not give more information than those that can be simply derived by checking your plots. Checking your Figure 2, it immediately appears that (as expected) you have more sea-salt at the coast than inland (leading to your PC1). Outside of that, other information derived from the PCA analysis are not very powerful (see my further comments).

**(15) author's response**

We agree that the spatial variations in ions on the traverse and the correlation plots of ions versus Na$^+$ (Figures 2 and 5 in the revised manuscript) can provide the information on main sources of ions in surface snow. For instance, the sea salt related ions show high concentrations on the coast. In this case, it seems that the PCA approach is redundant. As for the PCA approach, it can offer information on the geochemical behaviors of chemical ions in addition to the sources, i.e., PCA is a powerful tool for identifying the common sources and/or transport process of chemicals in different environments. Sometimes, chemicals with different sources can also be highly loaded in the same principle component, possibly due to the common geochemical behaviors (e.g., transport process). Therefore, the text associated with PCA was kept in the revised manuscript. But we made more use of the ion concentrations in excess with respect to the seawater composition when identifying the main sources of ions in snow, following the reviewer's comments.

**(15) author's changes in manuscript**

In the methodology section, the subsection on principal component analysis (PCA) of ions was kept in the manuscript. In section 4.2 Groups of ions in surface snow, the PCA results were shortened, and the main source identification of ions in surface snow were performed by making more use of the non-sea-salt fractions of ions.
Please see the revised manuscript, sections 2.3, 4.1, and 4.2.

**(16) comments from Referees**

Section 3.1:
Line 234-245: It is for a very short time that melted snow is under saturated with respect to

CO2. In fact after 10 min or so the equilibrium is reached but don't forget that another important factor is the temperature (colder is water more CO2 is dissolved). Another source of uncertainty here is the PCO2 in the lab of analysis (related for instance to the number of people). Please report the temperature at which your pH measurements were done.
Would be good here to compare your calculated H+ not only with previous similar estimates but also with previous actual measurements (Legrand and Delmas, 1985, for instance).

**(16) author's response**

Thanks for the helpful comments. We agree that temperature is an important parameter influencing the amount of $CO_2$ dissolved in the water, and the lower temperature is water and the more $CO_2$ is dissolved. Also, we agree that the number of people in the lab can influence the pH values potentially. In this study, the measurements of pH were performed in a class 100 clean room at room temperature ($\sim 20^oC$)

Following the reviewer's suggestion, the calculated $H^+$ concentrations in this study were also compared with previous direct measurements (Legrand and Delmas, 1985).

Legrand M. and Delmas R.J. (1985) Spatial and temporal variations of snow chemistry in Terre Adélie (East Antarctica). Ann. Glaciol. 7, 20-25.

**(16) author's changes in manuscript**

The measurement temperature was added in the revised manuscript, and the sentence reads,
"pH values of surface snow sampled in 2013 were measured with a glass pH electrode in a class 100 room at room temperature ($\sim 20^oC$),"
The calculated H+ concentrations were compared to the direct measurements in previous investigation, and a paragraph was added in the revised manuscript, as follows,
"Here, the calculated $H^+$ concentrations vary in the range of 0.51-10.01 μeq $L^{-1}$, with a mean of $3.53 \pm 1.61$ μeq $L^{-1}$. In general, the calculated $H^+$ values of the coastal surface snow are generally comparable to previous direct measurements in Terre Adelie (Legrand and Delmas, 1985).

**(17) comments from Referees**

Section 3.2:
Line 274: I feel that after having calculated nssSO4 (excess sulfate) you will identify an increasing trend of excess-sulfate from the coast to inland due to dry deposition (the sulfate one being obscured by the large amount of sea salt at the coast).

**(17) author's response**

In surface snow on the traverse, $nssSO_4^{2-}$, on average, accounts for more than 90% of the total $SO_4^{2-}$. That is, most of the $SO_4^{2-}$ in surface snow is from the marine biogenic emissions. The concentrations of $nssSO_4^{2-}$ in surface snow depend on the transport efficiency towards inland, the snow accumulation rate at a specific site, etc, and there was no significant correlation between $nssSO_4^{2-}$ concentrations and distance from coast on the traverse (Figure below).

[Figure]

Figure Correlation between $nssSO_4^{2-}$ concentrations in surface snow and distance from coast

**(17) author's changes in manuscript**

The spatial pattern of $nssSO_4^{2-}$ was included in the revised manuscript, as follows,

"As for $SO_4^{2-}$ (and $nssSO_4^{2-}$), $NH_4^+$, and $Ca^{2+}$, no clear spatial trend was found on the traverse."

**(18) comments from Referees**

Section 3.3:

This section needs to be significantly developed. First please show all species, second calculate excesses and corresponding error bars.

**(18) author's response**

Thanks for the constructive comment. Together with the comments from Reviewer#1, the Results section was re-organized and substantially revised. Section 3.3 in previous version was changed to section 3.1 (Chemical ion variations in snow pits) in the updated manuscript. In this section, concentrations of all species in snow pits (including $Cl^-$, $NO_3^-$, $SO_4^{2-}$($nssSO_4^{2-}$), $Na^+$, $NH_4^+$, $K^+$, $Mg^{2+}$ and $Ca^{2+}$) are presented. Then the non-sea-salt fractions of $K^+$, $Mg^{2+}$, and $Ca^{2+}$ are calculated (including the errors). Considering figure 3 (in revised manuscript) now contains the profiles of $Cl^-$, $NO_3^-$, $SO_4^{2-}$($nssSO_4^{2-}$), $Na^+$, $NH_4^+$, $K^+$, $Mg^{2+}$, $Ca^{2+}$, and the ratio of $SO_4^{2-}/Na^+$, the variations in $nssCl^-$, $nssK^+$, $nssMg^{2+}$, and $nssCa^{2+}$ were not included in Figure 3. (Note that $nssSO_4^{2-}$ and the ratio of $SO_4^{2-}/Na^+$ included in Figure 3 help to date the snow pits). A new paragraph was added in the revised manuscript to show the variations in non-sea-salt fractions of ions in the three snow pits, and the profiles of non-sea-salt fractions of ions ($nssCl^-$, $nssK^+$, $nssMg^{2+}$, and $nssCa^{2+}$) in snow pits P1 (a), P2 (b), and P3 (c) are now present in Figure S2 in the supplementary materials.

**(18) author's changes in manuscript**

This section (section 3.1 in the revised version) was substantially revised, and it now reads,

"Clear seasonal cycles of $Na^+$ and $nssSO_4^{2-}$ are present in P1 and P2, and thus the two pits can be well dated, spanning ~3 years (Figs. 3 (a) and (b)). Based on the snow pit dating, it is estimated that snow accumulation rate is ~50 (P1) and ~33 cm snow per year (P2), agreeing well with the field measurements (P1: ~150 kg m$^{-2}$ a$^{-1}$; P2: ~100 kg m$^{-2}$ a$^{-1}$; Fig. 2(a)), assuming a snow density of ~0.33 g cm$^{-3}$. At P1, negative $nssSO_4^{2-}$ values are observed in winter snow, i.e., $SO_4^{2-}/Na^+$ ratio below that of bulk seawater, while all of the $nssSO_4^{2-}$ data in P2 pit are positive. It is difficult to assign the samples in the snow pits to the four distinct seasons based on the measured parameters, and thus, in the following discussion, we choose a conservative assignment method, i.e., a summer season featured with higher $nssSO_4^{2-}$ and $SO_4^{2-}/Na^+$ ratio (and lower $Na^+$) and a winter season characterized with the opposite patterns. In addition to $SO_4^{2-}$ and $Na^+$, the other species also show seasonal variations, especially in pit P1, where elevated levels of $NO_3^-$ and $NH_4^+$ are generally present in summer snow, and the values of $Cl^-$, $K^+$, $Mg^{2+}$, and $Ca^{2+}$ are high in winter. It is noted that even in the same season, ion concentrations could vary among samples at a single site (e.g., shaded areas in Figs. 3(a) and (b)).

As for $nssSO_4^{2-}$ at P3, the very large signal at the depth of ~120 cm is most likely the fallout from the massive eruption of Pinatubo in 1991 (Fig. 3(c)), based upon previous observations at Dome A (e.g., Hou et al., 2007). Accordingly, the snow accumulation rate from 1992 to 2010 is ~22 kg m$^{-2}$ a$^{-1}$, in line with previous investigations (Hou et al., 2007; Jiang et al., 2012; Ding et al., 2016). Based on $nssSO_4^{2-}$ signals and the method proposed by Cole-Dai et al. (1997), 19 continuous samples have been identified as influenced by Pinatubo eruption, covering ~2.5 years, possibly suggesting that the effects of Pinatubo eruption on atmospheric chemistry lasted at least for 2.5 years over Dome A. Interestingly, only elevated $SO_4^{2-}$ concentrations are present during this period, and anomalous high or low concentrations of other ions are absent. Additionally, no correlation was found between $nssSO_4^{2-}$ and other species during the 2.5 years, possibly suggesting that Pinatubo volcanic emissions contribute less to the ion budgets other than $SO_4^{2-}$ at Dome A.

Previous investigations proposed that $Na^+$ and $nssSO_4^{2-}$ in surface snow (top ~1 cm) collected during a full year at central Antarctica show clear seasonal cycles, with high (low) $Na^+$ in winter (summer) snow (Udisti et al., 2012). At P3, $Na^+$, $nssSO_4^{2-}$ and the ratios of $SO_4^{2-}/Na^+$ fluctuate significantly (Fig. 3(c)), and these contrasts are unlikely indicative of the seasonal cycles as that for P1 and P2. In a full year of snow accumulation at P3, on average, about 7-8 samples were collected, allowing for examining the seasonal variability of ions. Following the field measurements of snow accumulation rate at Dome A during 2008-2011 (~20 kg m$^{-2}$ a$^{-1}$; Ding et al., 2015), the snow samples covering the years 2008 and 2009 can be roughly identified, assuming an even distribution of snow accumulation throughout the year. In total, there are 7 and 8 samples identified in the years 2008 and 2009, respectively (Fig. S1), and no seasonal cycles in $Na^+$, $nssSO_4^{2-}$, and $SO_4^{2-}/Na^+$ ratio were found due to the low snow accumulation rate at P3. In addition, the post-depositional processes (e.g., migration, diffusion, and ventilation processes) and/or wind scouring can obscure the original signal (Cunningham and Waddington, 1993; Albert and Shultz, 2002; Libois et al., 2014; Caiazzo et al., 2016), resulting in the absence of seasonal cycles of ions at P3.

In terms of the non-sea-salt fractions in the snow pits (Fig. S2), $nssCl^-$ is lower at P1 (0.25±0.28 μeq L$^{-1}$) than at the inland sites P2 and P3 (0.42±0.18 and 0.58±0.34 μeq L$^{-1}$, respectively), while the

concentrations of $nssK^+$, $nssMg^{2+}$, and $nssCa^{2+}$ generally show similar spatial patterns, possibly due to the low snow accumulation rate in interior areas. Different from the sea salt ions and $nssSO_4^{2-}$, $nssCl^-$, $nssK^+$, $nssMg^{2+}$, and $nssCa^{2+}$ in pits P1 and P2, do not show significant seasonal patterns. In the coastal pit P1, the non-sea-salt fractions account for less (<~30%) of the total ions, and the contribution percentages of non-sea-salt fractions increased at inland sites P2 and P3, about 30-70 %."

**(19) comments from Referees**

Section 4.1:
Please present excess here.
Line 371-373: I disagree with that since only a very small amount of potassium from dust is leachable (and measured with your IC) but it is not at all true for calcium (see Legrand et al., 1988, for instance). Therefore you cannot compare your snow data with the mean crust composition from Bowen that refers to total potassium (insoluble and soluble). Please check in Legrand et al. (1988) or Legrand (1987) information on excess potassium versus excess calcium.

**(19) author's response**

Thanks for the comments, and the concentrations in excess, instead of the EFs, are now presented.

We agree that the IC can only measure the soluble fractions of the terrestrial particle mass, and the insoluble fraction is largely unknown. The non-sea-salt fractions of ions (e.g., $nssMg^{2+}$ and $nssCa^{2+}$) can represent, at least partly, the terrestrial crustal materials. For instance, close relationship was found between $nssMg^{2+}$ or $nssCa^{2+}$ and aluminum concentrations at Dome C (Legrand and Delmas, 1988). We admit that possibly different portions of $K^+$ and $Ca^{2+}$ in the crustal material will be dissolved in the water (snow), and thus the calculation here seems to be uncertain regarding that Bowen's work is for the total crust composition (i.e., including both soluble and insoluble fractions). In this, we cannot get the ratio of $K^+/Ca^{2+}$ from the water soluble fractions of terrestrial particle mass, and thus the calculation in this section was removed and additional discussion on the sources of $K^+$ was included.

Legrand M.R. and Delmas R.J. (1988) Soluble impurities in four Antarctic ice cores over the last 30,000 years. Ann. Glaciol. 10, 116-120.

**(19) author's changes in manuscript**

The calculation of EFs was removed, and all of the EFs in the original version were replaced with the concentrations in excess (i.e., the non-sea-salt fractions).

The calculation of $K^+$ from the biomass burning emissions was removed, and additional discussion was included in the revised manuscript, as follows,
"Here, no elevated snow $nssK^+$ peaks were found in austral autumn (i.e., P1 and P2; Fig. S2) when the chemicals emitted from biomass burning (e.g., black carbon) often peaked in Antarctic snow (Sigl et al., 2016). In addition, there is a close relationship between $nssK^+$ and $nssCa^{2+}$ (r=0.22, p<0.001), suggesting that snow $nssK^+$ is unlikely dominated by biomass burning emissions."

**(20) comments from Referees**

I find also that your calcium data (Fig 2) are often above 4 ppb (that is also higher than seen in numerous ice cores under present-day climate. Please comment.

**(20) author's response**

Thanks for the comment. Indeed, values of $Ca^{2+}$ at some sites, especially on the coast and the in the inland regions (about 500-900 km from the coast). In the coastal areas, the high concentrations are likely associated with the marine inputs, while the elevated values at the inland sites could be related to the low and fluctuating snow accumulation rate due to the strong wind scouring (Ding et al., 2011; Das et al., 2013). In fact, similar elevated $Ca^{2+}$ values were found in the glaze/dune regions on the US ITASE traverses across East and West Antarctica (Dixon et al., 2013).

**(20) author's changes in manuscript**

A paragraph was included in the revised manuscript to explore the possible reasons for the high $Ca^{2+}$ concentrations at some sites, as follows,

"It is noted that some $Ca^{2+}$ concentrations in surface snow (e.g., coastal and some inland sites) are above 0.2 µeq $L^{-1}$(Fig. 2(i)), slightly higher than most reports of snow and ice under present-day climate (e.g., Legrand and Mayewski, 1997). On the coast, the high concentrations could be related to marine inputs (e.g., Bertler et al., 2005), while the elevated values in the inland regions (about 500-900 km from the coast, where the glaze/dune are distributed) are possibly associated with the low and fluctuating snow accumulation rate due to the strong wind scouring (Ding et al., 2011; Das et al., 2013). Similarly, in the glaze/dune regions on the US ITASE traverses across East and West Antarctica, concentrations of $Ca^{2+}$ in snow and ice are also often above ~0.2µeq $L^{-1}$ (Dixon et al., 2013)."

**(21) comments from Referees**

Line 445-454: I disagree with this discussion. Whereas I agree nitrate is clearly related to atmospheric oxidant, the link to sulfate (its similar presence in PC3) is due to the fact that you concentrate nssSO4 due to dry deposition at sites with low accumulation rate while nitrate is enhanced for a totally different reason (photochemistry).

**(21) author's response**

Thanks for the helpful comments. Indeed, high concentrations of both $nssSO_4^{2-}$ and $NO_3^-$ were found at the sites with low snow accumulation rate, while elevated $NO_3^-$ concentrations are on the Dome A plateau and high $nssSO_4^{2-}$ values occurred in the region about 300-800km from the coast (see figure below). Thus, snow accumulation rate is positively correlated with both species (see figure below). The positive loading of $SO_4^{2-}$ in PC3 and the correlation between $SO_4^{2-}$ and $NO_3^-$ in surface snow could be explained, at least in part, by the effects of snow accumulation rate.

[Figure]

Figure Annual snow accumulation rate, elevation (a) and concentrations of $NO_3^-$ (b) and $nssSO_4^{2-}$ (c) in surface snow collected during five seasons.

[Figure]

Figure Relationship between snow accumulation rate and $nssSO_4^{2-}$ (a) or $NO_3^-$ (b) in surface snow on the traverse.

In addition, $nssSO_4^2$ and $NO_3^-$ in the atmosphere are mainly associated with secondary aerosols, and the formation of both species is closely related to the oxidants $HO_x$, $RO_x$, etc (Ishino et al., 2017; Shi et al., 2018). Therefore, we cannot completely rule out that the correlation between $SO_4^{2-}$ and $NO_3^-$ may be associated with their formations in the atmosphere.

**(21) author's changes in manuscript**

Following the reviewer's comments, this point was re-discussed in the revised manuscript, as follows, "$SO_4^{2-}$ did not show high loadings in any of the three extracted components. Its positive loading in PC1 (0.55) and weak relationships between $SO_4^{2-}$ and sea salts (Cl⁻ and Na⁺) likely supports the contribution

of sea salt aerosols, although a minor one. A positive loading of $SO_4^{2-}$ is also present in PC3 (0.42), and a weak correlation was found between $SO_4^{2-}$ and $NO_3^-$. Both $SO_4^{2-}$ (or $nssSO_4^{2-}$) and $NO_3^-$ are negatively correlated with snow accumulation rate (Fig. 6), but with distinct mechanisms. $nssSO_4^{2-}$ can be concentrated due to dry deposition at sites with low snow accumulation rate, while elevated $NO_3^-$ concentrations are linked to the photochemical cycling and re-deposition (discussed above). In addition, $nssSO_4^{2}$ and $NO_3^-$ are mainly associated with the secondary aerosols, and the production of both species in summer is closely related to the oxidants $HO_x$, $RO_x$, etc (Ishino et al., 2017; Shi et al., 2018a), which may also contribute to the correlation between $SO_4^{2-}$ and $NO_3^-$."

**(22) comments from Referees**

Section 4.2:
Line 430: Your argument of an absence of correlation between ammonium and organic tracers (Shi et al., 2019) is not correct since the authors invoked a decrease of levoglucosan from the coast to inland due to its photochemical degradation.

**(22) author's response**

We agree that the levoglucosan could undergo photochemical degradation in the snow because of the 24-h sunlight during summertime in Antarctica. Thus the absence of correlation between $NH_4^+$ and levoglucosan could be associated with the post-degradation of levoglucosan. A recent observation on the same traverse suggested no correlation between $NH_4^+$ and black carbon that is mainly from Southern Hemisphere biomass burning emissions (Ma et al., 2020), in surface snow. In addition, no association was observed between $NH_4^+$ and phenolic compounds (i.e., vallinic and syringic acids, which are derived mostly from the combustion of lignin) in surface snow (Shi et al., 2019). In this case, the observations do not support that $NH_4^+$ is dominated by biomass burning emissions.

**(22) author's changes in manuscript**

This section was re-written following the comments from both reviewers, as follows,

"$Ca^{2+}$ is mainly from terrestrial particle mass, while $NH_4^+$ is thought to be mainly associated with biological decomposition of organic matter in the Southern Ocean (Johnson et al., 2007; Kaufmann et al., 2010). In addition, biomass burning from mid-latitudes can contribute to snow $NH_4^+$ at some sites (Pasteris et al., 2014), and the penguin colony emissions can be important inputs to $NH_4^+$ in snow several km from the colony (Rankin and Wolff, 2000). On this traverse, no correlation was found between $NH_4^+$ and biomass burning tracers (e.g., black carbon and phenolic compounds) in surface snow (Shi et al., 2019; Ma et al., 2020), suggesting a minor role of biomass burning emissions. Thus, the high $NH_4^+$ concentrations on the coast are likely associated with marine biogenic emissions. In this case, it is possible that a similar transport pathway (i.e., preferentially transported long distance in free transport; Krinner and Genthon, 2003; Kaufmann et al., 2010; Krinner et al., 2010) can explain, at least in part, the positive loadings of both $NH_4^+$ and $Ca^{2+}$ in PC2."

**(23) comments from Referees**

Line 433: Checking your Fig S1 in fact you have an outstanding value (17 ppb). Discarding this value I have difficulty to identify a seasonal cycle. Also your mean value (removing the outlier) is close to 2 ppb (it is slightly but significantly higher than what was seen in previous ice core studies). Please comment.

**(23) author's response**

 Thanks for the comment. Indeed, there is one sample with very high $NH_4^+$ concentration, ~0.9 μeq $L^{-1}$. If this data was excluded, there is no significant difference between the concentrations in summer and winter (Independent samples t test; $p>0.05$), although the mean concentration is slightly higher in summer than in winter. The discussion on the seasonal variation in $NH_4^+$ and Figure S1 (in original version) were removed in the revised manuscript.

 Indeed, the mean value of $NH_4^+$ at snow pit P1 is 0.16 μeq $L^{-1}$ (removing the outlier). The $NH_4^+$ concentrations here are slightly higher than what was observed in previous ice core studies, but generally comparable to some coastal observations (e.g., coastal sites in Terre Adelie) (Legrand and Delmas, 1985; Legrand et al., 1998). At the sites close to coastal line, especially near the penguin colony, snow $NH_4^+$ could be influenced by the penguin colony emissions (Legrand et al., 1998; Rankin and Wolff, 2000), but this effect is very localized, likely within several km from the colony (Rankin and Wolff, 2000). Considering that site P1 is 46 km from the coast and the ratio of $Na^+/K^+$ is close to that of bulk seawater (Figure 9 in the revised manuscript), snow $NH_4^+$ at P1 is unlikely dominated by local penguin colony emissions. Thus, $NH_4^+$ at the coastal site P1 is more likely influenced by marine biogenic emissions.

 Legrand M. and Delmas R.J. (1985) Spatial and temporal variations of snow chemistry in Terre Adélie (East Antarctica). Ann. Glaciol. 7, 20-25.
 Legrand M., Ducroz F., Wagenbach D., Mulvaney R. and Hall J. (1998) Ammonium in coastal Antarctic aerosol and snow: Role of polar ocean and penguin emissions. J. Geophys. Res. 103, 11043-11056.
 Rankin A.M. and Wolff E.W. (2000) Ammonium and potassium in snow around an emperor penguin colony. Antarct. Sci. 12, 154-159.

**(23) author's changes in manuscript**

The profile of $NH_4^+$ in P1 was included in Figure 3 in the revised manuscript, and the outstanding value was excluded. Also the relatively high concentrations of $NH_4^+$ in P1 was discussed, as follows,

 "$Ca^{2+}$ is mainly from terrestrial particle mass, while $NH_4^+$ is thought to be mainly associated with biological decomposition of organic matter in the Southern Ocean (Johnson et al., 2007; Kaufmann et al., 2010). In addition, biomass burning from mid-latitudes can contribute to snow $NH_4^+$ at some sites (Pasteris et al., 2014), and the penguin colony emissions can be important inputs to $NH_4^+$ in snow several km from the colony (Rankin and Wolff, 2000). On this traverse, no correlation was found between $NH_4^+$ and biomass burning tracers (e.g., black carbon and phenolic compounds) in surface snow (Shi et al., 2019; Ma et al., 2020), suggesting a minor role of biomass burning emissions. Thus,

the high $NH_4^+$ concentrations on the coast are likely associated with marine biogenic emissions. In this case, it is possible that a similar transport pathway (i.e., preferentially transported long distance in free transport; Krinner and Genthon, 2003; Kaufmann et al., 2010; Krinner et al., 2010) can explain, at least in part, the positive loadings of both $NH_4^+$ and $Ca^{2+}$ in PC2."

"In addition, $NH_4^+$ concentration at P1 ($0.16\pm0.05$ µeq $L^{-1}$) is slightly higher than the previous reports of ice cores but comparable to some coastal observations (e.g., coastal sites in Terre Adelie) (Legrand and Delmas, 1985; Legrand et al., 1998), possibly associated with marine biogenic emissions (i.e., close to the coast) (discussed above)."

**(24) comments from Referees**

Section 4.4: Given the poor representativeness of snow samples, it is clear that examination with respect to SOI is very difficult. For your information, based on a continuous record of 25 year of aerosol, Weller et al. (2011) examine the inter-annual variability with respect to climate-related indices (SAM, SOI, SIE) and nothing very significance had appear.
Weller R., Wagenbach D., Legrand M., Elsässer C., Tian-kunze X. and Königlanglo G., Continuous 25-years aerosol records at coastal Antarctica: I. Inter-annual variability of ionic compounds and links to climate indices, Tellus, 63B, 901-919, DOI:10.1111/j.1600-0889.2011.00542.x, 2011.

**(24) author's response**

We agree that the surface samples, which represent different lengths of time at varied locations, are unlikely representative of seasonal (or interannual) variations. Together with the comments of Reviewer#1, this section was removed.

Also, thanks for the reference, which provides much longer time of information. Indeed, it is difficult to construct the potential association between chemicals and the climate-related indices.

Weller R., Wagenbach D., Legrand M., Elsässer C., Tian-kunze X. and Königlanglo G. (2011) Continuous 25-yr aerosol records at coastal Antarctica: inter-annual variability of ionic compounds and links to climate indices. Tellus B 63, 901-919.

**(24) author's changes in manuscript**

This section was removed in the revised manuscript.

**References**

Bertler, N., Mayewski, P.A., Aristarain, A., Barrett, P., Becagli, S., Bernardo, R., Bo, S., Xiao, C., Curran, M., Qin, D., 2005. Snow chemistry across Antarctica. Ann. Glaciol. 41, 167-179.
Cole-Dai, J., Ferris, D.G., Lanciki, A.L., Savarino, J., Thiemens, M.H., Mcconnell, J.R., 2013. Two likely stratospheric volcanic eruptions in the 1450s C.E. found in a bipolar, subannually dated 800 year ice core record. J. Geophys. Res. 118, 7459-7466.

Das, I., Bell, R.E., Scambos, T.A., Wolovick, M., Creyts, T.T., Studinger, M., Frearson, N., Nicolas, J.P., Lenaerts, J.T., van den Broeke, M.R., 2013. Influence of persistent wind scour on the surface mass balance of Antarctica. Nat. Geosci. 6, 367-371.

Ding, M., Xiao, C., Li, Y., Ren, J., Hou, S., Jin, B., Sun, B., 2011. Spatial variability of surface mass balance along a traverse route from Zhongshan station to Dome A, Antarctica. J. Glaciol. 57, 658-666.

Dixon, D.A., Mayewski, P.A., Korotkikh, E., Sneed, S.B., Handley, M.J., Introne, D.S., Scambos, T.A., 2013. Variations in snow and firn chemistry along US ITASE traverses and the effect of surface glazing. Cryosphere 7, 515-535.

Ishino, S., Hattori, S., Savarino, J., Jourdain, B., Preunkert, S., Legrand, M., Caillon, N., Barbero, A., Kuribayashi, K., Yoshida, N., 2017. Seasonal variations of triple oxygen isotopic compositions of atmospheric sulfate, nitrate and ozone at Dumont d'Urville, coastal Antarctica. Atmos. Chem. Phys. 17, 1-25.

Jiang, S., Cole-Dai, J., Li, Y., Ferris, D.G., Ma, H., An, C., Shi, G., Sun, B., 2012. A detailed 2840 year record of explosive volcanism in a shallow ice core from Dome A, East Antarctica. J. Glaciol. 58, 65-75.

Johnson, M., Sanders, R., Avgoustidi, V., Lucas, M., Brown, L., Hansell, D., Moore, M., Gibb, S., Liss, P., Jickells, T., 2007. Ammonium accumulation during a silicate-limited diatom bloom indicates the potential for ammonia emission events. Mar. Chem. 106, 63-75.

Kaufmann, P., Fundel, F., Fischer, H., Bigler, M., Ruth, U., Udisti, R., Hansson, M., De Angelis, M., Barbante, C., Wolff, E.W., 2010. Ammonium and non-sea salt sulfate in the EPICA ice cores as indicator of biological activity in the Southern Ocean. Quaternary Sci. Rev. 29, 313-323.

Khodzher, T.V., Golobokova, L.P., Osipov, E.Y., Shibaev, Y.A., Lipenkov, V.Y., Osipova, O.P., Petit, J.R., 2014. Spatial-temporal dynamics of chemical composition of surface snow in East Antarctica along the Progress station-Vostok station transect. Cryosphere 8, 931–939.

Krinner, G., Genthon, C., 2003. Tropospheric transport of continental tracers towards Antarctica under varying climatic conditions. Tellus B 55, 54-70.

Krinner, G., Petit, J.R., Delmonte, B., 2010. Altitude of atmospheric tracer transport towards Antarctica inpresent and glacial climate. Quaternary Sci. Rev. 29, 274-284.

Legrand, M., 1987. Chemistry of Antarctic snow and ice. J. de Phys. 48, 77-86.

Legrand, M., 1995. Sulphur-derived species in polar ice: a review, in: Delmas, R. (Ed.), Ice core studies of global biogeochemical cycles. Springer, pp. 91-119.

Legrand, M., 1997. Ice–core records of atmospheric sulphur. Phil. Trans. R. Soc. Lond. B 352, 241-250.

Legrand, M., Delmas, R., 1987. A 220-year continuous record of volcanic $H_2SO_4$ in the Antarctic ice sheet. Nature 327, 671-676.

Legrand, M., Delmas, R.J., 1985. Spatial and temporal variations of snow chemistry in Terre Adélie (East Antarctica). Ann. Glaciol. 7, 20-25.

Legrand, M., Ducroz, F., Wagenbach, D., Mulvaney, R., Hall, J., 1998. Ammonium in coastal Antarctic aerosol and snow: Role of polar ocean and penguin emissions. J. Geophys. Res. 103, 11043-11056.

Legrand, M., Lorius, C., Barkov, N., Petrov, V., 1988. Vostok (Antarctica) ice core: Atmospheric chemistry changes over the last climatic cycle (160,000 years). Atmos. Environ. 22, 317-331.

Legrand, M., Wolff, E., Wagenbach, D., 1999. Antarctic aerosol and snowfall chemistry: implications for deep Antarctic ice-core chemistry. Ann. Glaciol. 29, 66-72.

Legrand, M.R., Delmas, R.J., 1984. The ionic balance of Antarctic snow: a 10-year detailed record. Atmos. Environ. 18, 1867-1874.

Ma, X., Li, C., Du, Z., Dou, T., Ding, M., Ming, J., Wang, M., Gao, S., Xiao, C., Wang, X., Ren, J., Kang, S., 2020. Spatial and temporal variations of refractory black carbon along the transect from Zhongshan Station to Dome A, eastern Antarctica. Atmos. Environ., 117816.

Mayewski, P.A., Carleton, A.M., Birkel, S.D., Dixon, D., Kurbatov, A.V., Korotkikh, E., McConnell, J., Curran, M., Cole-Dai, J., Jiang, S., Plummer, C., Vance, T., Maasch, K.A., Sneed, S.B., Handley, M., 2017. Ice core and climate reanalysis analogs to predict Antarctic and Southern Hemisphere climate changes. Quaternary Sci. Rev. 155, 50-66.

Mulvaney, R., Wolff, E., 1994. Spatial variability of the major chemistry of the Antarctic ice sheet. Ann. Glaciol. 20, 440-447.

Pasteris, D.R., McConnell, J.R., Das, S.B., Criscitiello, A.S., Evans, M.J., Maselli, O.J., Sigl, M., Layman, L.C.J.D., 2014. Seasonally resolved ice core records from West Antarctica indicate a sea ice source of sea-salt aerosol and a biomass burning source of ammonium. J. Geophys. Res. 119, 9168-9182.

Proposito, M., Becagli, S., Castellano, E., Flora, O., Genoni, L., Gragnani, R., Stenni, B., Traversi, R., Udisti, R., Frezzotti, M., 2002. Chemical and isotopic snow variability along the 1998 ITASE traverse from Terra Nova Bay to Dome C, East Antarctica. Ann. Glaciol. 35, 187-194.

Qin, D., Zeller, E.J., Dreschhoff, G.A., 1992. The distribution of nitrate content in the surface snow of the Antarctic Ice Sheet along the route of the 1990 International Trans-Antarctica Expedition. J. Geophys. Res. 97, 6277-6284.

Rankin, A.M., Wolff, E.W., 2000. Ammonium and potassium in snow around an emperor penguin colony. Antarct. Sci. 12, 154-159.

Russell, A., McGregor, G.R., 2010. Southern hemisphere atmospheric circulation: impacts on Antarctic climate and reconstructions from Antarctic ice core data. Climatic. Change 99, 155-192.

Severi, M., Becagli, S., Castellano, E., Morganti, A., Traversi, R., Udisti, R., 2009. Thirty years of snow deposition at Talos Dome (Northern Victoria Land, East Antarctica): Chemical profiles and climatic implications. Microchem. J. 92, 15-20.

Shi, G., Buffen, A.M., Ma, H., Hu, Z., Sun, B., Li, C., Yu, J., Ma, T., An, C., Jiang, S., Li, Y., Hastings, M.G., 2018. Distinguishing summertime atmospheric production of nitrate across the East Antarctic Ice Sheet. Geochim. Cosmochim. Acta 231, 1-14.

Shi, G., Wang, X.-C., Li, Y., Trengove, R., Hu, Z., Mi, M., Li, X., Yu, J., Hunter, B., He, T., 2019. Organic tracers from biomass burning in snow from the coast to the ice sheet summit of East Antarctica. Atmos. Environ. 201, 231-241.

Sigl, M., Fudge, T.J., Winstrup, M., Cole-Dai, J., Ferris, D., Mcconnell, J.R., Taylor, K.C., Welten, K.C., Woodruff, T.E., Adolphi, F., Bisiaux, M., Brook, E.J., Buizert, C., Caffee, M.W., Dunbar, N.W., Edwards, R., Geng, L., Iverson, N., Koffman, B., Layman, L., Maselli, O.J., McGwire, K., Muscheler, R., Nishiizumi, K., Pasteris, D.R., Rhodes, R.H., Sowers, T.A., 2016. The WAIS Divide deep ice core WD2014 chronology - Part 2: Annual-layer counting (0-31 ka BP). Clim. Past 12, 769-786.

Suzuki, T., Iizuka, Y., Matsuoka, K., Furukawa, T., Kamiyama, K., Watanabe, O., 2002. Distribution of sea salt components in snow cover along the traverse route from the coast to Dome Fuji station 1000 km inland at east Dronning Maud Land, Antarctica. Tellus B Chemical and Physical Meteorology 54, 407-411.

Weller, R., Wagenbach, D., Legrand, M., Elsässer, C., Tian-kunze, X., Königlanglo, G., 2011. Continuous 25-yr aerosol records at coastal Antarctica-: inter-annual variability of ionic compounds and links to climate indices. Tellus B 63, 901-919.

**End of responses to Prof. Michel Legrand.**

---

## Author Comment (AC4) · 7 Dec 2020

The comment was uploaded in the form of a supplement:
https://tc.copernicus.org/preprints/tc-2020-255/tc-2020-255-AC4-supplement.zip

---

## Author Comment (AC5) · 7 Dec 2020

The comment was uploaded in the form of a supplement:
https://tc.copernicus.org/preprints/tc-2020-255/tc-2020-255-AC5-supplement.zip

---

## Author Response (AR1)

**Editor Prof. Joel Savarino**

**We thank Prof. Savarino very much for his careful and thoughtful review of our work. Please see below for point-by-point responses in blue following Prof. Savarino's comments, in the order of (1) comments from Referees, (2) author's response.**
**Prof. Joel Savarino's comments are in black, and the responses are in blue.**

**(1) comments from Referees**

The reviewers still think that the propagation of errors is not correctly done and are not mentioned when certain estimates are made (eg lines 439, 473, 505) , that the scientific content is relatively poor although interesting questions are asked without the data being able to answer them and that the article is too long for its scientific content.

For example, the dechlorination of marine aerosols is a well-known process, so there is no need to write a 20+ lines paragraph to indicate that the results are consistent with previous studies;

**(1) author's response**

Thanks for the comments. In the revised manuscript, all of the errors were included when calculating the non-sea-salt fractions of ions in snow.

Following the comments from both editor and reviewer, the article was significantly shortened, and now it is a short communication instead of a research artic (see responses below).

Indeed, dechlorination of sea salt aerosols is a well-known process, and a detailed discussion on the process was omitted in the revised manuscript. In the updated version, only the possible source of nssCl⁻ in snow was simply mentioned. Please see the first paragraph in Section 4.1,

"nssCl⁻ accounted for $(39\pm24)$ % of Cl⁻ on the traverse, with higher values in the interior areas. The elevated fractions of nssCl⁻ are likely associated with the 'secondary' HCl which is produced by the reactions between sea salts and acids (e.g., $HNO_3$ and $H_2SO_4$) (Finlayson-Pitts, 2003)."

**(2) comments from Referees**

Similarly, the discussion on potassium is rather poor and does not allow to conclude, the ternary diagram is useless, a discussion on nss sulfate is sufficient, the principal components analysis does not bring any new element justifying such a development. Several essential references on e.g. the stability of nitrate in snow are missing (Freyer 1996, Neubauer 1988, Legrand 1990).

**(2) author's response**

Agreed and thanks for the comments.

For the discussion on main origins of nssK$^+$, the data of refractory black carbon (rBC), a tracer of biomass burning emissions in the Southern Hemisphere (Sigl et al., 2016), was included. In Antarctic snow, the nssK$^+$ has three main sources, biological activity on the coast (Rankin and Wolff, 2000), terrestrial particle mass transport, and combustion emissions in the Southern Hemisphere (Virkkula et al., 2006; Hara et al., 2013). Considering that all sampling sites are at least several tens of kilometers away from the coast, the contribution of biological activity to snow K$^+$ would be rather minor (Rankin and Wolff, 2000). A lack of correlation between K$^+$ (or nssK$^+$) and refractory black carbon (rBC; Figure below), which mainly represent the biomass burning emissions in the Southern Hemisphere (Sigl et al., 2016), suggests that K$^+$ is unlikely dominated by biomass burning emissions in surface snow.

[Figure]

**Figure.** Correlations between refractory black carbon (rBC) and K$^+$ (nssK$^+$) in Antarctic surface snow collected in 2012-2013 campaign. The rBC analysis system consists of an ultrasonic nebulizer and desolvation system (CETAC UT5000) coupled with a Single Particle Soot Photometer (SP2, Droplet Measurement Technologies, Boulder, Colorado).

Following the reviewer's suggestion, all ternary diagrams and related context were excluded in the revised manuscript. In the revised manuscript, only the non-sea-salt fractions of ions were discussed, and the discussion on the modification to sea salts with the aid of the relationships among Cl$^-$, Na$^+$ and SO$_4^{2-}$ was excluded.

Indeed, parts of the principal components analysis (PCA) results are similar to the outcomes of correlation analysis between ions and Na$^+$ in the original version. In this case, the parts associated with PCA method and discussions were excluded in the revised manuscript.

In the previous version, indeed, several import works on nitrate in snow were missing (Freyer 1996, Neubauer 1988, Legrand 1990). In the revised manuscript, the discussion on snow nitrate was excluded, considering the word limits of a brief communication and that nitrate in the snow across the Antarctic ice sheet (and on this traverse) has been extensively investigated (e.g., Frey et al., 2009; Erbland et al., 2013; Shi et al., 2015; Shi et al., 2018). Consequently, only the work of Neubauer and Heumann (1988) was included in the introduction in the revised manuscript.

Neubauer, J., and Heumann, K.G.: Nitrate trace determinations in snow and firn core samples of ice shelves at the Weddell Sea, Antarctica, Atmospheric Environment (1967), 22, 537-545, 1988.

**(3) comments from Referees**

The correlations of nitrate and sulfate with accumulation have already been shown and either their interpretation has already been made (sulfate) or do not allow to deduce a result without knowing a priori this result by other means (nitrate).

**(3) author's response**

Agreed. Following the comments, the correlations of chloride, nitrate, and sulfate with snow accumulation rate were excluded in the revised manuscript. In the manuscript, we only focus on the occurrence, sources and variability of non-sea-salt fractions of ions in snow. Please see the revised manuscript.

**(4) comments from Referees**

In spite of this and like the reviewers, I think that the data from the traverse deserves to be published, especially because this type of measurement is rare in Antarctica. I therefore suggest to the authors to drastically shorten the size of the manuscript by focusing on the new results of their study compared to previous ones and passing quickly over the similarities. To this end I recommend the submission of a short paper in the form of a brief communication rather than a research article.

**(4) author's response**

Thanks to Prof Joel Savarino, and following your suggestion, the scientific content in the manuscript was significantly shortened. Now, the revised version is in the form of a brief communication rather than a research article, and we only focus on the non-sea-salt fractions of ions in the revised manuscript. Several important revisions were made to meet the requirements of a brief communication:
1) The manuscript title now is, Brief communication: Spatial and temporal variations in surface snow chemistry along a traverse from coastal East Antarctica to the ice sheet summit (Dome A)
2) The abstract was significantly shortened and improved. Now, the total word number is ~100, meeting the requirements of a brief communication.
3) Three figures (required by a brief communication) were kept in the revised version. The ternary diagram, correlations of chloride, nitrate, and sulfate with snow accumulation rate, the PCA results are omitted. The other figures were moved to the supplementary materials.
4) The Introduction was significantly shortened, and spatial and temporal variations in chemical ions in Antarctic snow and ice were generally reviewed. Then, raised why we carried out the present investigation.
5) In the section Results, the ion variations in snow pits and ion concentrations in surface snow were concisely summarized.

6) In the section Discussions, we focus on: 1) the non-sea-salt fractions of ions in surface snow, including the spatial patterns and main sources of the non-sea-salt fractions of ions, and 2) non-sea-salt fractions and fluxes of ions in snow pits, including the seasonal variations of the non-sea-salt fractions of ions and the related controlling factors, and the ion fluxes were also calculated.

7) The section Conclusions was revised accordingly.

Through the abovementioned revisions, the updated version tends to meet a brief accumulation.

**References**

Erbland, J., Vicars, W., Savarino, J., Morin, S., Frey, M., Frosini, D., Vince, E., and Martins, J.: Air-snow transfer of nitrate on the East Antarctic Plateau - Part 1: Isotopic evidence for a photolytically driven dynamic equilibrium in summer, Atmos. Chem. Phys., 13, 6403-6419, doi:10.5194/acp-13-6403-2013, 2013.

Finlayson-Pitts, B.J.: The tropospheric chemistry of sea salt: a molecular-level view of the chemistry of NaCl and NaBr, Chem. Rev., 103, 4801-4822, 2003.

Frey, M.M., Savarino, J., Morin, S., Erbland, J., and Martins, J.: Photolysis imprint in the nitrate stable isotope signal in snow and atmosphere of East Antarctica and implications for reactive nitrogen cycling, Atmos. Chem. Phys., 9, 8681-8696, 2009.

Hara, K., Osada, K., and Yamanouchi, T.: Tethered balloon-borne aerosol measurements: seasonal and vertical variations of aerosol constituents over Syowa Station, Antarctica, Atmos. Chem. Phys., 13, 9119-9139, 2013.

Rankin, A.M., and Wolff, E.W.: Ammonium and potassium in snow around an emperor penguin colony, Antarct. Sci., 12, 154-159, doi:10.1017/S0954102000000201, 2000.

Shi, G., Buffen, A.M., Hastings, M.G., Li, C., Ma, H., Li, Y., Sun, B., An, C., and Jiang, S.: Investigation of post-depositional processing of nitrate in East Antarctic snow: isotopic constraints on photolytic loss, re-oxidation, and source inputs, Atmos. Chem. Phys., 15, 9435–9453, doi:10.5194/acp-15-9435-2015, 2015.

Shi, G., Hastings, M.G., Yu, J., Ma, T., Hu, Z., An, C., Li, C., Ma, H., Jiang, S., and Li, Y.: Nitrate deposition and preservation in the snowpack along a traverse from coast to the ice sheet summit (Dome A) in East Antarctica, The Cryosphere, 12, 1177–1194, doi:10.5194/tc-12-1177-2018, 2018.

Sigl, M., Fudge, T.J., Winstrup, M., Cole-Dai, J., Ferris, D., Mcconnell, J.R., Taylor, K.C., Welten, K.C., Woodruff, T.E., Adolphi, F., Bisiaux, M., Brook, E.J., Buizert, C., Caffee, M.W., Dunbar, N.W., Edwards, R., Geng, L., Iverson, N., Koffman, B., Layman, L., Maselli, O.J., McGwire, K., Muscheler, R., Nishiizumi, K., Pasteris, D.R., Rhodes, R.H., and Sowers, T.A.: The WAIS Divide deep ice core WD2014 chronology - Part 2: Annual-layer counting (0-31 ka BP), Clim. Past, 12, 769-786, doi:10.5194/cp-12-769-2016, 2016.

Virkkula, A., Teinilä K., Hillamo, R., Kerminen, V.M., Saarikoski, S., Aurela, M., Koponen, I.K., and Kulmala, M.: Chemical size distributions of boundary layer aerosol over the Atlantic Ocean and at an Antarctic site, Atmos. Chem. Phys., 6, 303-310, 2006.

**End of responses to Prof. Savarino.**